# Intraseasonal to interannual variability of Kelvin wave momentum fluxes as derived from high-resolution radiosonde data

Jeremiah P. Sjoberg[1,2], Thomas Birner[1], and Richard H. Johnson[1]

[1]Department of Atmospheric Science, Colorado State University, Fort Collins, CO 80521
[2]now at: COSMIC Project Office, University Corporation for Atmospheric Research, Boulder, Colorado

*Correspondence to:* Jeremiah P. Sjoberg (sjoberg@ucar.edu)

**Abstract.** Observational estimates of Kelvin wave momentum fluxes in the tropical lower stratosphere remain challenging. Here we extend a method based on linear wave theory to estimate daily time series of these momentum fluxes from high-resolution radiosonde data. Daily time series are produced for sounding sites operated by the U.S. Department of Energy (DOE) and from the recent Dynamics of the Madden-Julian Oscillation (DYNAMO) field campaign. Our momentum flux estimates are found to be robust to different data sources and processing, and in quantitative agreement with estimates from prior studies. Testing the sensitivity to vertical resolution, our estimated momentum fluxes are found to be most sensitive to vertical resolution greater than 1 km, largely due to overestimation of the vertical wavelength. Climatological analysis is performed over a selected 11 year span of data from DOE Atmospheric Radiation Measurement (ARM) radiosonde sites. Analyses of this 11-year span of data reveal the expected seasonal cycle of momentum flux maxima in boreal winter and minima in boreal summer, and variability associated with the quasi-biennial oscillation of maxima during easterly phase and minima during westerly phase. Comparison between periods with active convection that is either strongly or weakly associated with the Madden-Julian Oscillation (MJO) suggests that the MJO provides a nontrivial increase in the lowermost stratospheric momentum fluxes.

## 1  Introduction

Atmospheric equatorial Kelvin waves represent a tropical eastward propagating wave disturbance generated primarily by convection. As tropical convection is nearly ubiquitous, particularly near to the Intertropical Convergence Zone, Kelvin waves are regular phenomena. These equatorially trapped waves have zero meridional wind perturbations and are a consequence of the equatorial beta-effect (Gill, 1982). Convectively coupled Kelvin waves are important for tropospheric phenomena, such as the Madden-Julian Oscillation (MJO) and the El Niño-Southern Oscillation (ENSO) (Straub et al., 2006; Kiladis et al., 2009). These interactions arise from the eastward emanation of energy associated with these waves. Energy is also emitted upwards by Kelvin waves into the tropical tropopause layer (TTL). In particular, these waves transport westerly momentum from the troposphere into the TTL and above.

Due to this upward flux of westerly momentum, Kelvin waves are known to influence the downward progression of stratospheric westerlies that occur during the roughly 28 month cycle of winds known as the quasi-bienniel oscillation (QBO). There

are established connections between the QBO and midwinter polar stratospheric variability (Garfinkel et al., 2012), large-scale extratropical weather patterns (Thompson et al., 2002), and stratospheric chemistry (Randel and Wu, 1996). Prior studies have shown that the QBO affects the vertical extent of convection (Collimore et al., 2003), and stratospheric water vapor through modulation of the cold-point temperature (Fueglistaler and Haynes, 2005).

Phasing of the QBO is initiated in the upper stratosphere by gravity waves of opposite propagation direction to that of the winds (e.g. for westerly winds, westward-propagating gravity waves drive reversal to easterlies; Baldwin et al., 2001). In the middle to lowermost stratosphere, additional waves come into play in each phase: Rossby waves and mixed Rossby-gravity waves during westerly phases and Kelvin waves during easterly phases. For a given state of the zonal winds (e.g. easterly), waves with phase velocity in the direction of the mean flow (e.g. Rossby waves) encounter a critical level in the

lower stratosphere and are inhibited from propagating. Meanwhile, waves with phase speed in the opposite direction to the mean flow (e.g. Kelvin waves) are free to propagate through the stratosphere until they dissipate or encounter a critical level, depositing their momentum. After sufficient momentum deposition forces the background winds to reverse direction, the waves that forced the new wind state are no longer able to propagate, allowing the other regime of waves to continue the oscillation. See Baldwin et al. (2001) for a thorough review of QBO theory and impacts.

A missing component from our understanding of the interaction between Kelvin waves and the QBO is a precise measure of the actual vertical transport of zonal momentum. This transport quantity is most typically written as the eddy momentum flux $\overline{\rho u'w'}$, the density-weighted zonal mean of the product between zonal mean deviations of zonal wind $u$ and vertical wind $w$. In principle, quantification of this flux only requires knowledge of the zonal distribution of these two fields. Measuring zonal wind does not present any considerable challenges as it has both typical speeds that are well above instrument sensitivity and

large zonal coherence. Vertical wind, in contrast, does not display either of these characteristics: typical vertical wind speeds are on the order of 1 cm s$^{-1}$ and zonal variations are large. While some observational platforms – such as flux towers – may reasonably estimate the vertical winds, they are spatially inhomogeneously distributed and only measure within the boundary layer.

Observational shortcomings such as these do not prevent estimation of the Kelvin wave momentum fluxes, however. Maruyama

(1968), through judicious application of dynamical theory, showed that the covariance of zonal wind and vertical wind may be approximated by the quadrature spectrum of zonal wind and temperature. That is, the out-of-phase relationship between these two readily observable fields may be used to estimate the vertical momentum fluxes by Kelvin waves. Later studies based on radiosonde data showed that these waves account for a non-negligible portion of the westerly momentum when the QBO is in its easterly phase (e.g. Sato et al., 1994; Maruyama, 1994). Other theory-based estimates of momentum fluxes have been de-

rived for use with satellite irradiances (Hitchman and Leovy, 1988; Ern and Preusse, 2009) and have found similar magnitudes of the momentum fluxes. Together with estimations using reanalysis products (e.g. Tindall et al., 2006), qualitatively consistent bounds to the momentum flux amplitudes have been determined.

Yet there remain places where our understanding and estimation may be improved. For instance, few climatological analyses of Kelvin wave momentum fluxes have been performed. While satellite and reanalysis studies have long data records over which

to analyze, the vertical resolution of both data sources is greater than a kilometer in the lower stratosphere. It is not clear how

sensitive momentum flux calculations are to vertical resolution, particularly for lower stratospheric Kelvin waves with vertical wavelengths on the order of 2-4 km. Kim and Chun (2015) showed that such waves may be significantly under-resolved in reanalyses. Studies using high vertical resolution radiosonde data meanwhile only analyzed broad characteristics of Kelvin waves during easterly and westerly QBO phases (e.g. Sato and Dunkerton, 1997), or point estimations of the flux amplitudes (e.g. Holton et al., 2001).

Here we extend previous methods using radiosonde data to analyze climatologies, variabilities, and vertical resolution dependences of Kelvin wave momentum fluxes. We make use of both quality-controlled, high resolution data from a recent field campaign and raw, high resolution data from long-term radiosonde stations. We apply an algorithm for producing continuous, daily time series of momentum flux estimates. By varying the vertical stepping of input data to the algorithm, we determine the role of vertical resolution on estimations of the fluxes. Because we utilize a relatively long-term dataset, we are able to analyze the intraseasonal and interannual variability of these time series of momentum fluxes, demonstrating that our methodology reproduces expected qualitative structures. From our estimates of the flux, we find evidence that a positive contribution to the momentum flux variability is provided by convection associated with the MJO.

Section 2 describes the data and methods we use to generate our flux estimates. Vertical resolution dependence is discussed in Section 3. We present time series of our results in Section 4. The annual mean and QBO-mean climatologies are discussed in Section 5. We discuss implications of these results in Section 6 and summarize in Section 7.

## 2 Data and methods

### 2.1 Data

The radiosonde data we use come from two sources. The first source contains short-term but high resolution data from the Dynamics of the Madden-Julian Oscillation (DYNAMO) field campaign (Yoneyama et al., 2013). One objective of DYNAMO was to analyze initiation of the MJO, in part through collection of frequent high-resolution radiosonde data. We use Level 4 radiosonde data (see Ciesielski et al., 2014) from the Gan Island (0.7°S, 73.2°E) and Manus Island (2.0°S, 147.4°E) sounding sites. These Level 4 data are produced at 50 m vertical resolution and 3-hourly temporal resolution. The Gan Island data span from 22 September 2011 to 08 February 2012, while the Manus Island data span from 24 September 2011 to 30 March 2012.

The second radiosonde source contains lower temporal resolution but longer spanning data from two U.S. Department of Energy Atmospheric Radiation Measurment (ARM) program sounding sites: Manus Island and Nauru Island (0.5°S, 166.9°E). See Mather and Voyles (2013) for a review of ARM facility instruments and missions. Data are recorded every 2 seconds, corresponding to a vertical step of roughly 10 m, by the sounding packages on launches with frequency ranging from once daily to 8 times daily during intensive operation periods such as DYNAMO. The number of data early within the record are insufficient for the spectral filtering we apply so we only consider data from 2003 January 01 to 2013 December 31 for this study. Sondes were launched at least twice daily during this 11 year range.

Data from the European Centre for Medium Range Weather Forecasting (ECMWF) Interim Reanalysis (ERAi) are also used (Dee et al., 2011) for estimating momentum fluxes and for zonal mean zonal winds. The data are on a regular 0.75° grid in

longitude and latitude, and are given at six-hourly temporal resolution. Model level data are used here to take advantage of the highest available vertical resolution in the TTL (∼1000-1500 m).

Outgoing longwave radiation (OLR) data are from a long-term record of daily, 1° x 1° gridded observations retrieved primarily by the high-resolution infrared radiation sounder instruments on board the National Oceanic and Atmospheric Administration (NOAA) TIROS-N series and Eumetsat MetOp polar orbiting satellites. These data, developed by a joint partnership between NOAA and the University of Maryland are obtained from the NOAA National Centers for Environmental Information (https://www.ncdc.noaa.gov/news/new-outgoing-longwave-radiation-climate-data-record).

For analyzing the MJO, we use the OLR MJO Index (OMI) index (Kiladis et al., 2014). OMI is based on the principal component time series of the first two empirical orthogonal functions of 20°S-20°N, 20-96 day filtered OLR. OMI data are obtained from the Physical Sciences Division of the NOAA Earth System Research Laboratory (https://www.esrl.noaa.gov/psd/mjo/mjoindex/).

## 2.2 Methods

To grid the radiosonde data, raw data are linearly interpolated in height and cubic spline interpolated in time. To constrain the interpolation, we require that each output data point has at least 3 input data points within the span from 3 days prior to 3 days following, and at least 3 input data points within the span from 500 m above to 500 m below. Note that this interpolation does not fill all gaps, allowing for missing data to remain. The results are not significantly different for other orders of interpolation, nor for changes in the time range or spatial range in which data points must exist in order to interpolate to a specified output point. This does not hold if the time range or spatial range is too small – shorter than 1 day or less than 100 m, respectively – in which case few output points will be produced.

We range the output temporal resolution from 6 hours to 48 hours, and the output vertical resolution from 100 m to 2000 m. These ranges are used to study resolution effects on the calculated momentum fluxes. For our standard analysis, we use temporal resolution of 24 hours and vertical resolution of 250 m. While daily data are standard for the field, motivation for why we use vertical stepping of 250 m is given in the next section.

To estimate the Kelvin wave momentum fluxes, we follow the technique described in Sato and Dunkerton (1997) and Holton et al. (2001). Wave solutions to the linearized equatorial beta-plane equations result in

$$\rho \overline{u'w'} = -\rho \frac{R\omega_d}{HN^2} Q_{uT}, \tag{1}$$

where $R$ is the gas constant; $H$ is the scale height of the atmosphere, taken to be 7 km here[1]; $N^2$ is the squared Brunt-Väisälä frequency; $Q_{uT}$ is the quadrature spectrum between $u$ and temperature $T$, given in units of K m s$^{-1}$; and $\omega_d$ is the intrinsic frequency. Inclusion of density $\rho$ in Eq. (1) casts the momentum flux as a stress term (in units of mPa) of which the vertical gradient approximates the Transformed Eulerian Mean wind forcing

$$\partial_t \overline{u} \propto -\rho^{-1} \partial_z \left( \rho \overline{u'w'} \right). \tag{2}$$

---

[1]Although this scale height is too long for the lower stratosphere, it is the value used in the literature (e.g. Andrews et al., 1987). Hence, we use 7 km to be consistent with past studies.

Following Andrews et al. (1987), with no meridional wind perturbations $v'$ and the WKBJ assumption, the intrinsic frequency takes the form

$$\omega_d \equiv \omega - k\overline{u} = -\frac{kN}{m}. \tag{3}$$

Here, $k$ is the zonal wavenumber satisfying $k = 2\pi/L_x$, where $L_x$ is the zonal wavelength of the Kelvin waves. $m$ is the vertical wavenumber and is defined to be negative. In deriving the intrinsic frequency, we assume that both the zonal mean zonal wind and stratification $N^2$ vary in the vertical and in time, but that variations are slow relative to variations in the phase of the waves. While this condition is generally true in time and space, variations in zonal wind can be large such that our estimates of the flux near, say, regions of QBO wind transition are more uncertain. WKBJ approximations, and thus the applied wave solutions used in deriving Eq. (1), may not be applicable in these regions. But as shown later in Section 4, we find good agreement between our results and those from previous studies, lending credibility to our estimates.

For stratification, this slowly-varying assumption is true above the tropopause inversion layer (TIL, e.g. Grise et al., 2010), and we find that 18 km typically lies above the TIL. We thus set the lower boundary of our analyses to this level. For the upper boundary of our analyses, 30 km is a natural choice as few radiosondes reach or extend beyond this level. To highlight results in the lowermost statosphere, we use 25 km as an upper boundary for the presented figures.

The values of $\omega$, i.e. the frequencies relative to the ground, are determined by the frequency bands of the spectral transform we apply to the sounding data. These bands are determined by the time stepping of our data and by the windowing of our spectral decomposition – here taken to be 40 days. This window length allows us to calculate the momentum fluxes at the principal Kelvin wave periods that lie between 5 and 20 days. For the 40 day windowing, the central periods are 20, 13.3, 10, 8, 6.7, 5.7 and 5 days.

The zonal means of both zonal wind and temperature are approximated by their time mean over each window. Such an approximation is reasonable in the stratosphere. We have compared our estimates of the zonal mean fields to those calculated from ERAi and found that the fields do not qualitatively differ. From our approximations of $\overline{u}$ and $\overline{T}$, we calculate the zonal deviation fields $u'$ and $T'$, and the stratification $N^2$.

Vertical wavenumbers $m$ are estimated by modifying a method described in Holton et al. (2001). For each data window, we filter $u'$ and $T'$ for each frequency band using a Butterworth bandpass filter. Then for each of the 40 days within each data window, we find the longest vertically contiguous span of data between 15 and 30 km for both fields at all temporal frequencies. 15 km is used as it represents the upper boundary of convection (e.g., Feng et al., 2014; Xu and Rutledge, 2015) from which Kelvin waves will emanate. The vertical quadrature spectrum $Q_{uT,v}$ is calculated from these contiguous spans of data. From the vertical wavelengths that have phasing such that temperature leads zonal wind, we select the wavelength with the largest negative value of $Q_{uT,v}$. The phase difference between temperature and zonal wind is required to be 45°-135° so as to avoid ambiguity for phase differences close to 0° and 180°. These steps result in an estimate of $L_z$ for each of the 40 days in a given data window and for each temporal frequency band. We use the window-mean $L_z$ to then calculate the vertical wavenumber $m = -2\pi/L_z$.

This method of estimating the vertical wavenumber was applied to the Nauru99 data that Holton et al. (2001) analyzed. Their estimate of 4.5 km for wave periods of 9-11 days is close to ours of 4.0 km for the same wave periods. Likewise, our estimated range of 3.2-5.1 km vertical wavelength for wave periods of 4-6 days is very similar to the range of 3-4.5 km by Holton et al. (2001).

With $m$ known, we may reorder Eq. (3) to obtain the relation for zonal wavenumber $k$:

$$k = 2\pi/L_x = \omega / \left( \overline{u} - \frac{N}{m} \right) . \tag{4}$$

For positive $\omega$ and sufficiently strong easterlies, Eq. (4) results in negative $k$. Since $k$ is defined to be positive for Kelvin waves, such regions – most often found in strong easterlies critical to QBO evolution – must be excluded from our analysis. But, there is no sign constraint imposed on $\omega$. We make use of this and of the symmetry of spectral amplitudes about positive

and negative $\omega$, i.e. the spectral amplitude $\hat{\chi}$ for a given field $\chi$ has the following characteristic: $\hat{\chi}(\omega) = \hat{\chi}(-\omega)$. Since we know the spectral amplitudes for positive $\omega$, we simply select the sign of $\omega$ such that $k$ is positive from Eq. (4). This however allows the estimated values of $L_x$ to be small ($\ll 100$ km) where we flip the sign of $\omega$. Since short zonal wavelength Kelvin waves are indistinguishable from gravity waves (Matsuno, 1966), we constrain our calculations to points where the horizontal wavelengths are greater than 500 km. We also require that the intrinsic zonal phase speed $\omega_d/k$ be positive, a fundamental

requirement for Kelvin waves.

Anywhere the above constraints are violated, we set the momentum fluxes to be missing. In order to restrict the impact of these eliminated regions on the analysis presented below, we only retain those points for which every frequency band has real data. This is perhaps more restrictive than is necessary, but ensures consistency, particularly for climatological analyses.

An additional constraint we could apply is that the meridional wind perturbations $v'$ be negligible. This is an important

distinguishing property between Kelvin waves and inertio-gravity waves. Appropriately implementing this constraint here is imprecise, however. Unlike with zonal wind, the assumption that the time mean approximates the zonal mean of meridional wind is not accurate. Our estimates of $v'$ will not then be representative of wave perturbations from the zonal mean. Nevertheless, we have tested only retaining data points that fulfill $|v'| < 5$ m s$^{-1}$ and found that our results are not significantly altered by that. Hence, we refrain from applying this constraint.

With all inputs known for Eq. (1), we calculate the momentum flux for each temporal frequency band between 5 and 20 days. Our method of calculation is performed at consecutive windows spaced 1 day apart. We then concatenate all these data windows to produce a daily time series of total Kelvin wave momentum fluxes. For this time series, the dates of the momentum fluxes are set to the middle date of each window. This technique of calculating spectral amplitudes in overlapping windows is more commonly referred to as a short-time Fourier transform (STFT). The use of STFT differs from prior work which used

non-overlapping windows of data to estimate momentum flux. As a result, we have an effecitvely higher temporal resolution, allowing for greater detail in analyses of intraseasonal and interannual variability of the estimated flux.

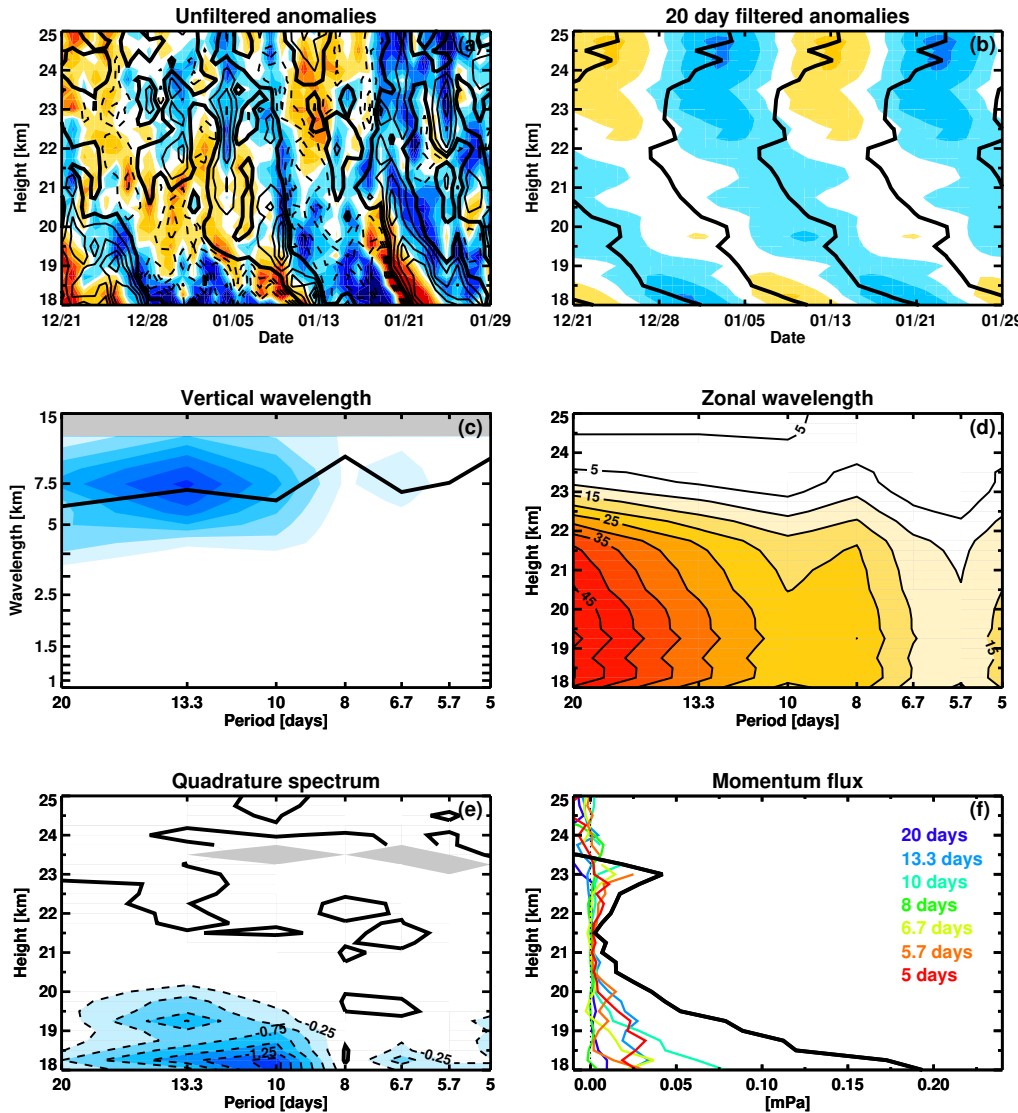

**Figure 1.** An example of the Kelvin wave momentum flux calculation for Gan Island over the 40 day window between 21 December 2011 and 29 January 2012. (a) Unfiltered zonal temperature anomalies (shading) and zonal wind anomalies (contours). The zonal wind and temperature contour intervals are 3 m s$^{-1}$ and 1 K, respectively. (b) 16-27 day bandpass filtered temperature and zonal wind over the data window. These correspond to the 20 day period waves in our calculations. (c) 40-day mean vertical quadrature spectrum between u$'$ and T$'$ (shading) as a function of period and vertical wavelength, and calculated vertical wavelength (black curve, units km) as a function of period. Dark blue shading indicates large magnitudes of the quadrature spectrum. (d) Zonal wavelength as a function of height and of period. Units are 1000 km. (e) The temporal quadrature spectrum between u$'$ and T$'$. Units are K m s$^{-1}$. (f) The calculated total 5-20 day momentum flux (black curve) as a function of height. The colored curves show the contributions from individual periods.

## 2.3 Example

An example of our method of calculation is shown in Fig. 1. The data for this example are from Gan Island and cover the 40 day window between 21 December 2011 and 29 January 2012. Unfiltered temperature anomalies (shading) and zonal wind anomalies (contours) are shown in panel (a). The estimated zonal mean zonal wind for this data window may be seen in the top panel of Fig. 3. The signature of Kelvin waves is present here: temperature anomalies lead zonal wind anomalies of the same sign with a roughly $90°$ phase difference. To better highlight this, panel (b) shows 16-27 day (20 day central period) bandpass filtered temperature and zonal wind. From these filtered anomalies, we expect to find large Kelvin wave momentum fluxes in the region between 18 and 21 km.

Panel (c) shows the mean vertical quadrature spectrum as a function of vertical wavelength and frequency in shading. The associated estimate of the vertical wavelengths is overplotted by the black curve. For this window, the waves we analyze have vertical wavelengths between 5-10 km, which fall well within observed ranges of vertical wavelengths for equatorial Kelvin waves (e.g., Randel and Wu, 2005). Zonal wavelengths are shown in (d) as a function of height and of wave period. The longest waves are at the longest periods and primarily below 21 km.

Panel (e) shows the quadrature spectrum between zonal wind anomalies and temperature anomalies as a function of period and of height. These spectra are used for calculation of the momentum fluxes shown in panel (f). As expected, the total momentum flux is large between 18 and 20.5 km with a roughly linear decrease in magnitude throughout this vertical span. Above this, the momentum flux is nearly constant with only moderate variations up to 25 km. The calculated fluxes in (f) are assigned to 09 January 2012, the central date of the window. Note that the wavenumbers and frequencies derived from the results shown in Fig. 1 are consistent with Kelvin waves (cf. Fig. 3 of Wheeler and Kiladis, 1999).

## 3 Resolution dependence

We test the dependence of the calculated momentum flux amplitudes on the resolution of input data by independently varying the vertical and temporal resolutions of the imposed interpolation. These tests are performed for a reference level of 18 km. We carry out the following tests at levels above 18 km and find that the results qualitatively hold. At higher levels, a number of complicating effects may come into play. First, coexistence of strong easterlies and relatively short vertical wavelengths occurs frequently above 20 km. As these two conditions tend to result in horizontal wavelengths shorter than 500 km, such regions violate the assumptions that allow the derivation and application of Eqs. (1) and (3). Second, with increasing height, data gaps increase as balloon bursts limit the maximum height of radiosondes. By analyzing just above the tropopause, we are more likely to minimize the impacts from these complications.

Fig. 2 (a) shows, as a function of vertical resolution, the mean 5-20 day Kelvin wave momentum fluxes over these common points (solid black). Values of the momentum flux are given on the right axis while percent differences from the reference calculation using 250 m vertical resolution are given on the left axis. The percent difference is defined here as $PD = (M - M_0)/((M + M0)/2) * 100$, where $M$ is the value at each vertical resolution, and $M_0$ is the reference value. For this analysis, we only use those data points that are not missing in any of the tested resolutions (total of 3240 points).

There is a strong linear relationship between the vertical resolution and the percent differences of the calculated momentum fluxes, with a mean increase of ~1.2% for every 100 m increase in vertical stepping. The standard errors of each measurement are shown with error bars for each tested resolution, where the number of input points has been reduced by the e-folding time scale of the momentum flux data (23 days). The difference between the mean flux at 250 m and those at both 1800 m and 2000

5  m are significantly different at the 90% level.

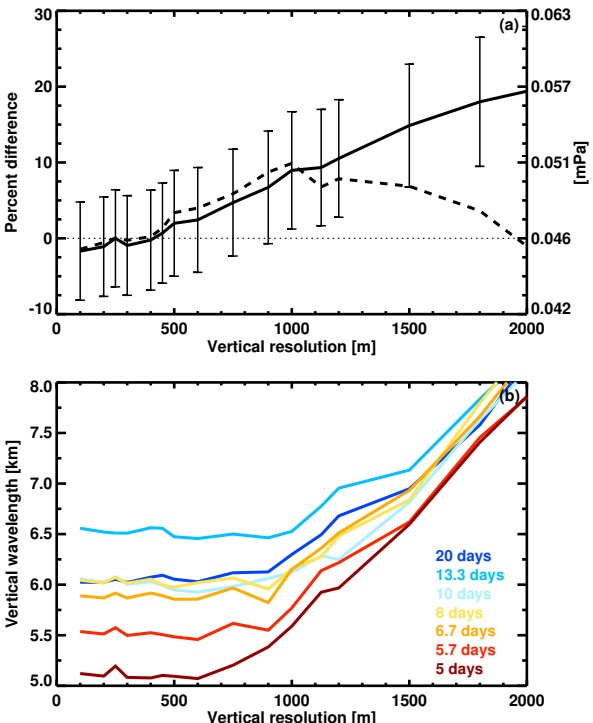

**Figure 2.** (a) Percent difference in the momentum fluxes from 250 m vertical resolution (solid). The right axis gives the values of the flux. The dashed curve shows values for calculations done with $L_z$ set to the 250 m resolution values. Standard errors are shown with bars for each tested resolution. (b) The variation of vertical length scale as a function of vertical resolution, shown here for each analyzed wave period.

We also study the impact of holding individual input parameters constant at the 250 m resolution values. The inputs considered here are stratification $N^2$, horizontal length scale $L_x$, vertical length scale $L_z$, and quadrature spectrum $Q_{uT}$. Holding $N^2$, $L_x$, or $Q_{uT}$ steady is found to not strongly influence the calculated fluxes: the momentum fluxes still increase for increasing vertical resolution at roughly the same rate (not shown).

10  In contrast, for constant vertical wavelength (dashed), the calculated fluxes increase to 900 m resolution and then decrease for still larger vertical stepping. This suggests that a large portion of the dependence of momentum fluxes on vertical resolution greater than 1000 m results from changes in the vertical wavelength. Panel (b) shows the how the vertical wavelength varies as

a function of vertical resolution. Up to ∼1000 m resolution, the vertical wavelength is insensitive to vertical resolution. From 1000 m to 2000 m resolution, the mean vertical wavelength at all wave periods linearly increases to be greater than 7.5 km. Notably, the minimum vertical wavelengths (not shown) have an analogous dependence: below 1000 m vertical stepping, the wavelength minima are all approximately 2 km, whereas these minima necessarily increase to be greater than 5 km by 2000 m
stepping.

Performing analytic experiments with Eqs. (1)-(4) verify that overestimation of the vertical wavelengths leads to overestimation of the momentum flux if the background wind is westerly. In easterlies, the effect is reversed but of relatively smaller magnitude, i.e., a given overestimation of vertical wavelength in westerlies increases the flux more than it is decreased in easterlies. Note that this relationship implies that the mean zonal wind for the points analyzed in Fig. 2 are westerly (see Fig.
10  6).

A similar analysis for changing temporal resolution finds that the overall variations from changes in time stepping are much smaller than those from vertical resolution (not shown). For time steps between 0.25 to 2 days, the mean flux is at most 5% different from the value for 24 hour resolution.

## 4   Time series results

Fig. 3 shows time series of 5-20 day Kelvin wave momentum fluxes calculated from DYNAMO data collected at Gan Island (top) and Manus Island (bottom). Both time series capture the same qualitative structure: positive fluxes in October and November 2011, followed by a large, sustained burst of positive flux that begins in early January 2012. The vertical extents of these momentum fluxes in both time series are also comparable, with amplitudes in both cases negligible above ∼21 km. This agreement is verified through linear correlation coefficients between the two time series that are between 0.67 and 0.91 at all
levels from 18 through 20 km. Lagged correlations do not suggest that there is a time offset between the two time series.

A noteworthy difference between the two is that the amplitudes at Gan Island are roughly 2-3 times larger than those at Manus Island. While this is certainly the case for the month of January, it is perhaps more obvious for the period between mid-October and mid-November. This amplitude difference arises in part because our method estimates the momentum fluxes from a point source of data. While Gan Island is located in a region of the Indian Ocean that is relatively far removed from
other land surfaces, Manus island is located just to the east of the Maritime Continent. The Maritime Continent is known to diminish both Kelvin waves (Flannaghan and Fueglistaler, 2012) and convective signals – such as the MJO (Zhang, 2005) – that force these waves.

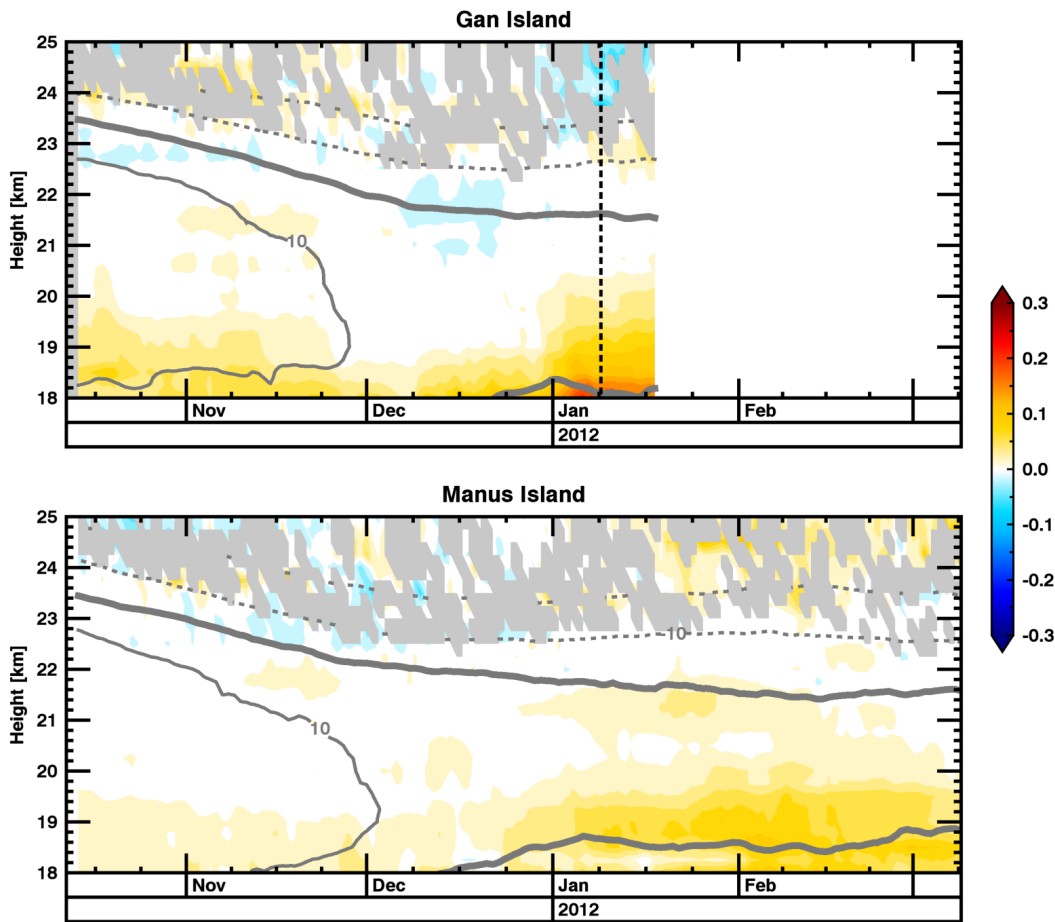

**Figure 3.** Time series of 5-20 day Kelvin wave momentum fluxes (shading) for Gan Island (top) and Manus Island (bottom) over the DYNAMO field campaign. The dark gray contours give the estimated zonal mean zonal wind at each site; solid contours are westerlies while dashed contours are easterlies. The 0 m s$^{-1}$ contour is shown in bold gray. The vertical black dashed line denotes the central date of the window used in Fig. 1. Light gray shading indicates where the momentum flux is not being calculated for any frequency band. Momentum flux is in units of mPa and zonal wind contour spacing is 10 m s$^{-1}$.

Figs. 4 and 5 show the time series of momentum fluxes for the Manus ARM site. We find that there is strong qualitative and quantiative agreement between momentum fluxes calculated from both the Manus and Nauru ARM sites. See supplementary Figs. S1 and S2 for the time series of momentum fluxes from the Nauru ARM site.

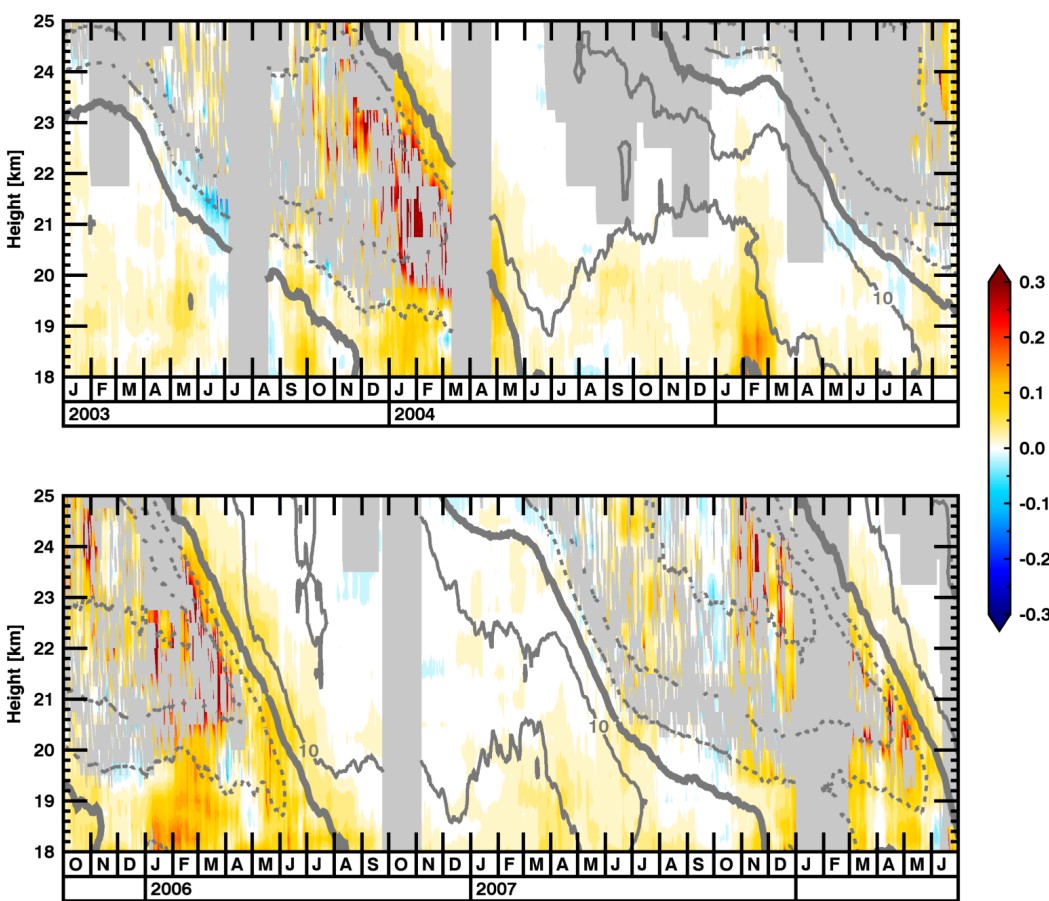

**Figure 4.** Time series of momentum fluxes from the Manus ARM site. Plotted fields are as in Fig. 3. The span here covers 01 January 2003 through 30 June 2008. Momentum flux is in units of mPa and zonal wind contour spacing (gray contours) is 10 m s$^{-1}$.

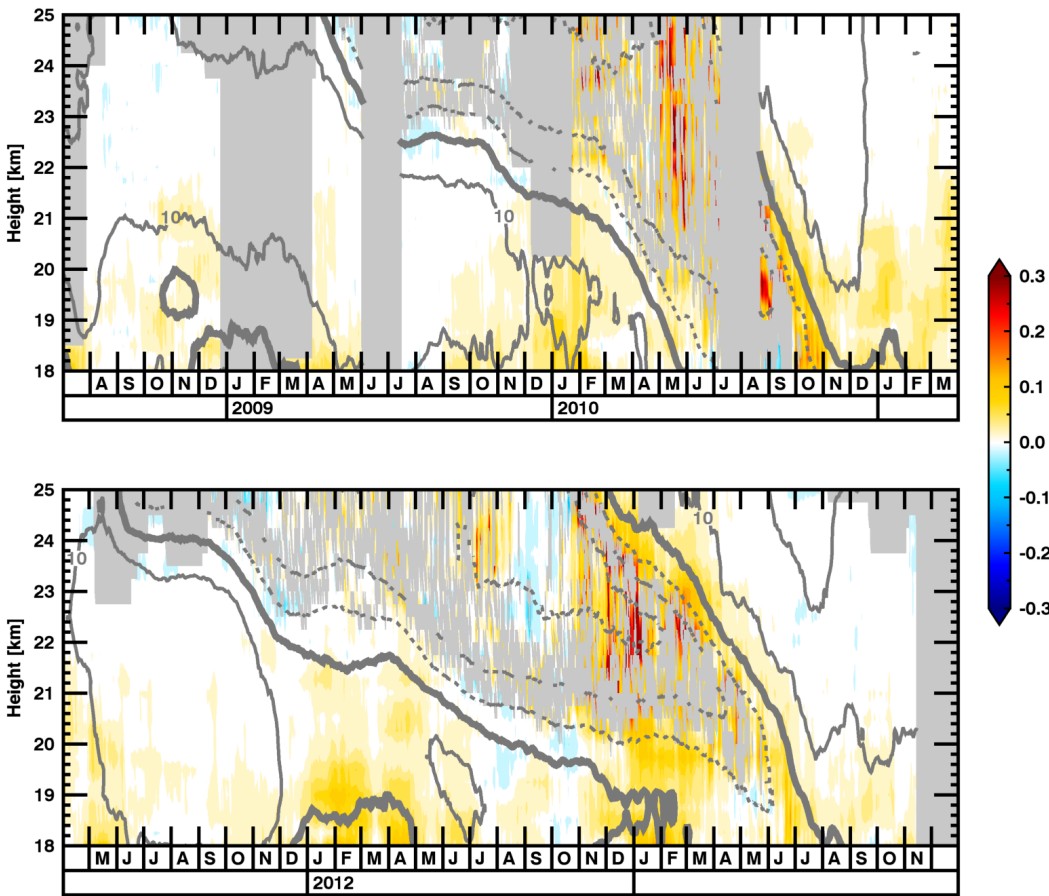

**Figure 5.** Time series of momentum fluxes from the Manus ARM site, continued. The span here covers 01 July 2008 through 31 December 2013.

In the time series, there are broad regions of data in the center of the easterlies in which the theoretical assumptions are violated (gray shading, see discussion in Section 2); these regions are excluded. There are also spans and heights with missing radiosonde data that are identifiable by the rectangular regions of gray shading. Where the data are not missing in easterlies, the qualitative patterns of the fluxes match expectations: Kelvin waves are influencing the descent of the westerlies by fluxing westerly momentum there. This is most clearly seen through descending maxima of the flux amplitudes that track the 0 m s$^{-1}$ contour. In westerly QBO phases, these fluxes are nonzero but much smaller.

We find that our estimated momentum flux amplitudes qualitatively agree with those estimated by Ern and Preusse (2009, their Figs. 4 and 5), though with somewhat smaller amplitudes. For example, the momentum flux increase between 18-20 km for JFM in 2006 (bottom panel of Fig. 4) is identified in Ern and Preusse (2009), but their amplitude reaches at least 0.3 mPa

while ours is at most ~0.20 mPa. A similar comparison with momentum fluxes derived from ERAi (not shown) also finds qualitative agreement with our amplitudes being smaller.

Calculated momentum fluxes for the Manus ARM site are remarkably similar to those from the Manus DYNAMO data, when the resolution between the two datasets is the same (cf. Fig. 3 and the lower panel of Fig. 5). The linear fit of these two datasets has a correlation coefficient of 0.94 and an offset less than 0.0006 mPa. Fidelity between momentum fluxes from both data sources is encouraging, as the long record length of the ARM data greatly increases the time over which we can analyze the waves.

One interpretation of the above results that arises when comparing simultaneous calculations of momentum fluxes at different sounding sites (see Figs. 3 and 5) is as follows: our technique for calculating momentum fluxes is strongly influenced by the local conditions. For the DYNAMO data, Gan Island observations occur within the Indian Ocean where strong convection associated with the Madden-Julian Oscillation (MJO) is common. The expansive convection associated with these MJO events experienced at Gan Island propagates eastward to Manus Island, but is diminished after propagating over the Maritime Continent (Zhang, 2005). One would expect that such a disruption to the convection will lead to a disruption in the generation of Kelvin waves. This may, in part, explain why the estimated momentum flux amplitudes are smaller at Manus Island, located to the east of the Maritime Continent. Fluxes at Nauru, ~20° to the east of Manus, are similarly smaller (see Figs. S1 and S2).

Our calculated momentum fluxes largely represent local contributions to the zonal mean. However, stratospheric Kelvin waves are known to strongly project onto plantery scales (i.e. zonal wavenumbers 1-3, Feng et al., 2007). This projection may be in part from a quasi-stationary source of these waves: frequent convection over the Indian Ocean and Maritime Continent. It is possible that data from Indian Ocean sounding sites routinely sample the amplitude maxima of these planetary-scale wave momentum fluxes. Our results may then give an upper estimate of the flux amplitudes. Future, more detailed analyses may reveal more insight into this issue.

## 5   Annual cycle and the QBO

The annual cycle of momentum fluxes and of input fields to the calculation of Eq. (1) for Manus Island are shown in Fig. 6. These mean fields are smoothed with a 21-day boxcar window for ease of viewing. In this and all remaining analyses involving averaging or compositing, we omit any points that have missing data. Note that this results in undersampling of the easterly winds in, e.g., Fig. 6 (a) since there are many missing data in this phase of the QBO (see Figs. 4 and 5).

Panel (a) shows that the lower stratospheric momentum fluxes maximize in January through March (JFM) and minimize in July through September. Each of the input fields to Eq. (1) shows variations that reinforce the variations in momentum flux (panels b-d). Particularly in JFM, the zonal wavelengths become shorter, and the quadrature spectrum amplitude maximizes. Panel (a) additionally shows that, over our data record, despite the westerly phase persisting longer (see Figs. 4 and 5), the easterly phase of the QBO has stronger zonal winds above 21 km than those during the westerly phase.

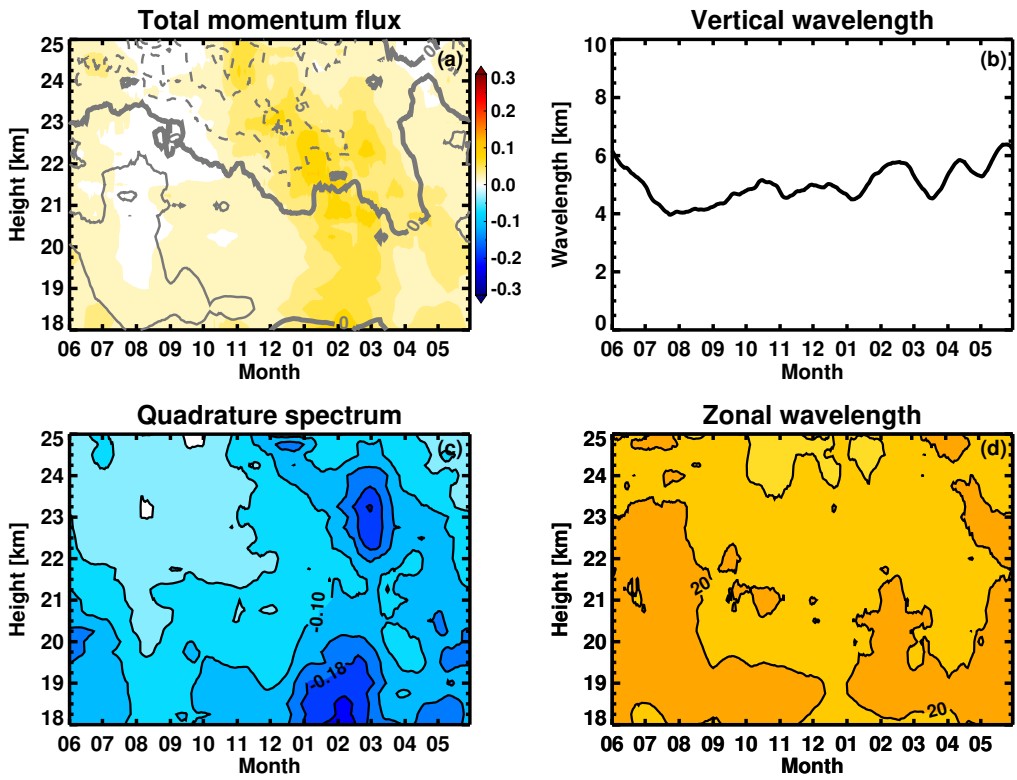

**Figure 6.** Climatology of momentum flux and related fields from the Manus ARM sounding site. (a) The 5-20 day total momentum flux [mPa] in shading and zonal mean zonal wind in contours. (b) The vertical length scale $L_z$ [km]. (c) The quadrature spectrum $Q_{uT}$ [K m s$^{-1}$]. (d) The horizontal length scale $L_x$ [1000 km] with contour spacing of 5000 km. The vertical wavelength, quadrature spectrum, and zonal wavelength are means over all wave periods and all fields are smoothed with a 21-day boxcar filter.

We next analyze the typical momentum fluxes associated with the QBO. We determine phases of the QBO based on an index of the phasing between zonal mean zonal wind at 30 hPa ($U_{30}$) and at 50 hPa ($U_{50}$). Note that our method is qualitatively similar to, though simpler than, the EOF-based index described in Wallace et al. (1993). This index is calculated as

$$QBO_i = \frac{1}{\pi} \tan^{-1}\left(\frac{U_{50}}{U_{30}}\right) ,$$ (5)

5 and thus ranges from -1 to +1, increasing in time until it reaches +1 and then restarting at -1. $QBO_i$ values between -1 and 0 broadly signify the transition period from easterlies to westerlies throughout the stratosphere, while values between 0 and +1 broadly signify the westerly to easterly transition. By firstly smoothing the zonal winds at both levels with a 31-day boxcar window, this produces a nearly monotonic, cyclic index of the state of the QBO. For this study, we use zonal wind data from ERAi and we split this index into 8 phases with steps of 0.25.

**Table 1.** Average residence time and standard errors, in units of days, for each bin of $QBO_i$. Only the four full cycles of the QBO in our data are analyzed here.

| Bin center | Residence time |
|---|---|
| -0.875 | 163 $\pm$29.2 |
| -0.625 | 40.3 $\pm$5.15 |
| -0.375 | 55.4 $\pm$3.62 |
| -0.125 | 35.0 $\pm$4.56 |
| 0.125 | 175 $\pm$56.3 |
| 0.375 | 99.5 $\pm$35.2 |
| 0.625 | 83.8 $\pm$27.1 |
| 0.875 | 168 $\pm$54.9 |

Fig. 7 shows the composite zonal mean zonal wind about this phasing index in gray contours. This composite structure aptly reproduces the expected zonal wind structure of the QBO. For the given 11 years of data, there are four full cycles, defined here as $QBO_i$ -1 to +1. These four cycles range in length from $\sim$24.0 months to $\sim$33.7 months, with a mean length of $\sim$27.3 months. The means and standard errors of the residence time for each bin is given in Table 1. The range and mean here are well within the observed values (e.g. Baldwin et al., 2001), indicating that this is an appropriate index onto which to composite our calculated momentum fluxes.

The composite momentum flux fields are also shown in Fig. 7. The largest Kelvin wave momentum flux occurs in the core of the easterlies while the largest vertical gradient in the flux occurs during the transition from easterlies to westerlies, as expected (Holton and Lindzen, 1972). These maxima track downwards with the transition until the lowermost stratospheric wind is westerly. That this descent occurs primarily during DJFM (Fig. 8) explains why the annual mean climatology displays descending signals of the flux and of easterlies during this span. Once the lowermost stratospheric winds are westerly, momentum fluxes become small except within the range of 18-20 km. This is consistent with prior observations of Kelvin waves during westerly QBO phase (Das and Pan, 2013).

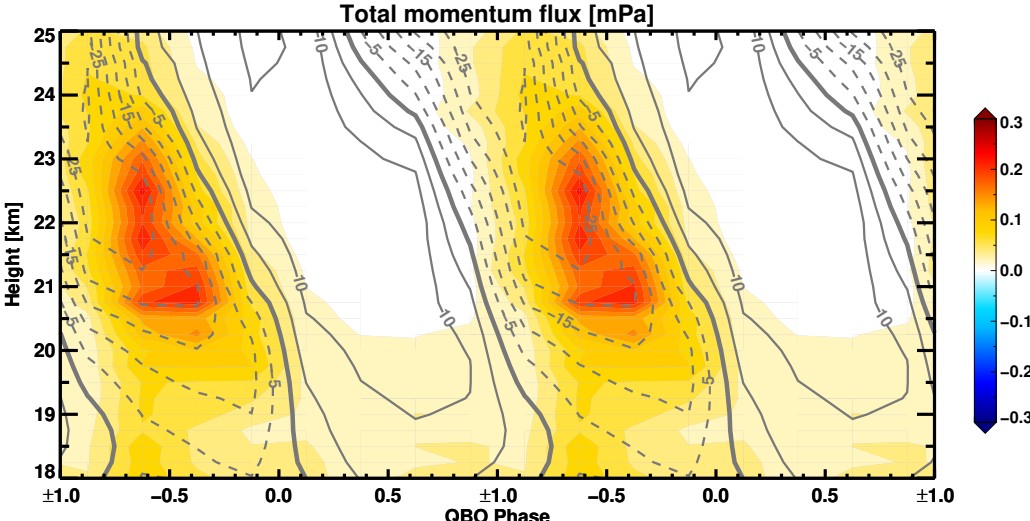

**Figure 7.** Composite momentum flux in shading and composite zonal mean zonal winds in gray contours, both as functions of $QBO_i$ (see text for details). Zonal wind contour spacing is $5 \text{ m s}^{-1}$ and the $0 \text{ m s}^{-1}$ contour is bolded. Two full cycles of the composite QBO are shown here to ease visual inspection.

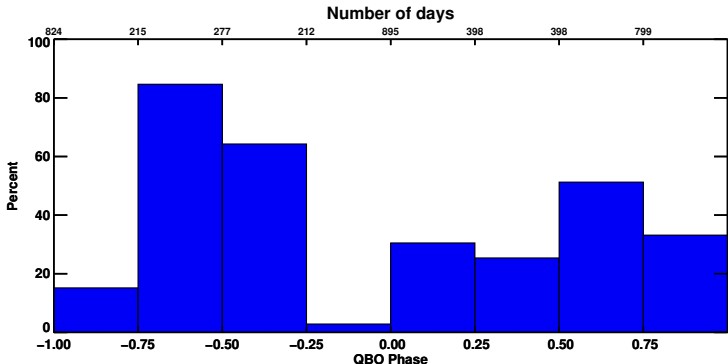

**Figure 8.** Percent of days in each QBO phase bin that fall between December and March. The total number of days in each bin is given along the top abcissa.

## 6  Discussion: organized convection and Kelvin waves

A natural step in this line of study is to analyze the relation between tropical convection and our estimated Kelvin wave momentum fluxes. However, tying a certain stratospheric Kelvin wave packet to a specific tropospheric deep convection event

is not necessarily straightforward. While Kelvin waves are predominantly generated by convection, not all tropical convection will generate Kelvin waves. When Kelvin waves are forced, their upward propagation will be strongly affected – and often inhibited – by changes in background winds and static stability near the tropical tropopause (e.g., Ryu et al., 2008; Flannaghan and Fueglistaler, 2012, 2013). Those Kelvin waves that do enter the stratosphere decouple from source convection (Kiladis et al., 2009) making direct attribution difficult without advanced methods for tracking these wave packets.

Rather than relying on advanced techniques to thoroughly quantify the effects of tropical deep convection on stratospheric Kelvin waves, here we show a cursory analysis of this relationship through use of compositing (superposed epoch analysis). Compositing relies on an appropriate definition of an event start date in order to filter out noise in the data, leaving the desired signal. Fundamentally, this method shows the lead-lag relationship between two fields: in our case between a measure of deep convective activity and stratospheric Kelvin waves.

Selecting an event in tropical convection is not straightforward since it is a nearly ubiquitous feature there. However, tropical deep convection is not simply spatially and temporally stochastic, but instead routinely organizes into large-scale patterns, most notably the Madden Julian Oscillation (MJO, Zhang, 2005). And since Kelvin waves project onto the largest scales – zonal wavenumber 1-3 and periods longer than 5 days (Feng et al., 2007; Scherllin-Pirscher et al., 2017) – a criterion that accounts for the large-scale organization of convection would be appropriate.

Here we define an event as any continuous span of days during which outgoing longwave radiation (OLR) indicative of deep convection covers more than $66\%$ of the upstream area for a given sounding site. We use 200 W/m$^2$ as our indicator of deep convection in OLR; 180 W/m$^2$ was also used and found to not result in qualitative changes to the results. The $66\%$ coverage threshold represents the 90th percentile value of coverage for our reference upstream area. The reference upstream area – taken here to be the $30°$ longitude west of and $\pm 2.5°$ latitude around each location – is analyzed because our estimated momentum fluxes are strongly influenced by the environment near to the sounding site (see Section 4). We note that the following results qualitatively hold for other configurations of the upstream area, including if all points between $10°$S-$10°$N are used, so long as the coverage threshold is set to be the 90th percentile value for the considered upstream area. Individual events last for as long as the coverage threshold is continuously met or exceeded. Since the threshold is set to a high level, we do not require a minimum duration. To better isolate the Kelvin wave signals, we do require events to be separated by at least 10 days. Longer separation lengths lead to substantially fewer identified events, but the results qualitatively hold.

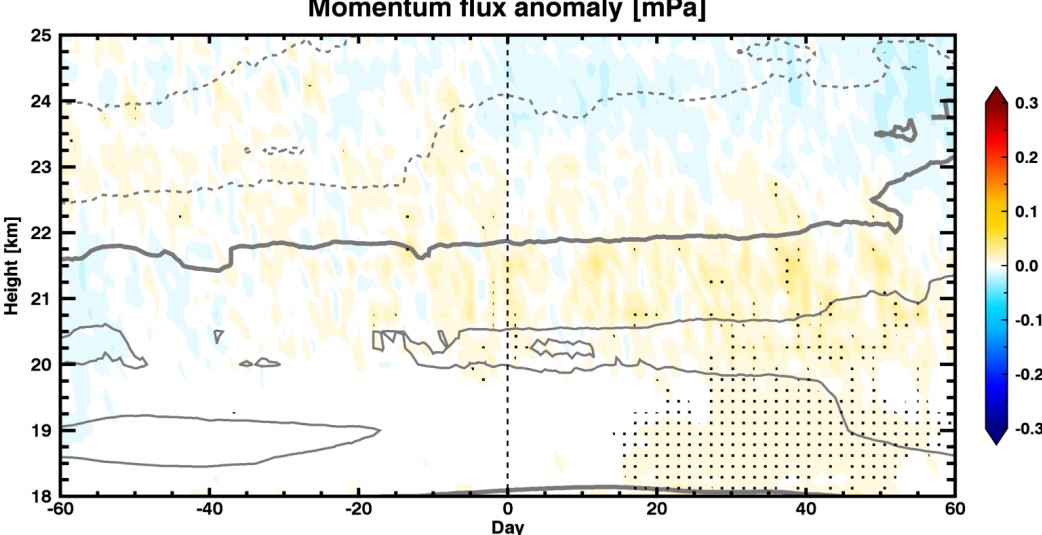

**Figure 9.** Composite momentum flux anomaly for events where OLR is less than 200 W/m$^2$ over at least 66% of the reference upsteam area for Manus Island. Day 0 (dashed black line) is the first day this condition is satisfied. The anomaly is the difference from the mean over the entire data record. Stippling indicates where the anomaly is significant at the 95% level. Gray curves are the composite zonal mean zonal wind at 2.5 m s$^{-1}$ spacing, with the 0 m s$^{-1}$ contour bolded. See the text for more details.

Using these event identification criteria, we find 88 events for the Manus sounding site. The composite Kelvin wave momentum flux anomaly and zonal wind are shown in Fig. 9. The anomaly is the difference from the time mean over all points. To evaluate the significance at the 95% level of our composite values, we use a 10000-member Monte Carlo simulation to calculate the confidence interval of a background (randomly selected) composite (Laken and Čalogović, 2013). There is a sig-
5  nificant signal of positive flux into the lowermost stratosphere in the 20-50 day range following our events. The amplitude of this positive flux represents a ∼0.50-0.75 standard deviation anomaly. A similar pattern of variability is found in lag regressions between momentum flux and our convective coverage data (not shown), though the correlations are at most 0.3.

The relatively small amplitudes of the anomalies and low linear correlations indicate that spatially-dense signals in OLR are not the primary factor preceding positive anomalies of momentum flux in the stratosphere. For instance, this composite analysis
10  does not account for the background wind state or changes in stratification, and is only for a single radiosonde site that may miss some or all of the momentum flux signal generated by the local convection. Yet, the 95% significant signal shown here is in line with expectations from theory: organized convection leads to generation of Kelvin waves. These results are also in line with those from Randel and Wu (2005), who showed the broad relationship between Indonesian OLR and lower stratospheric Kelvin wave temperature variance.

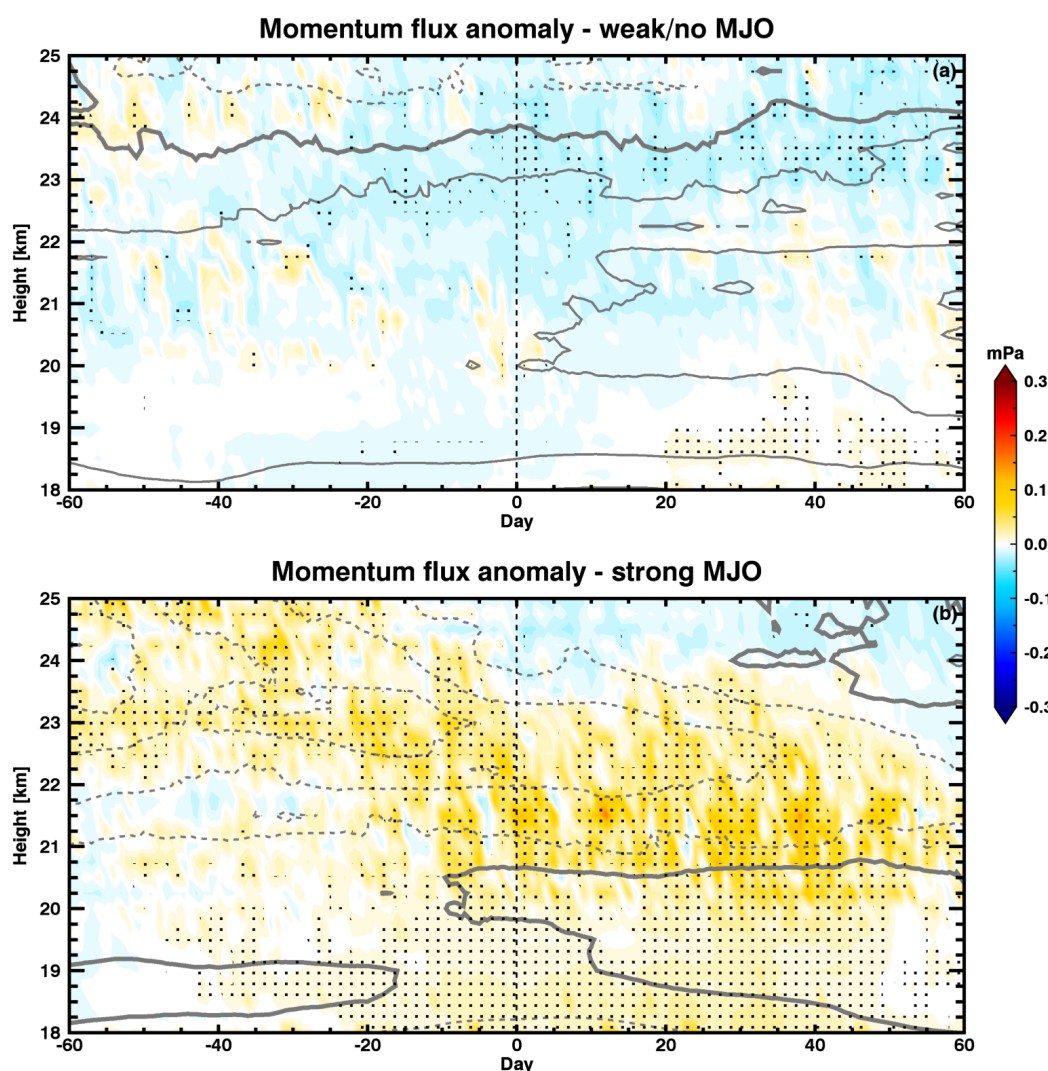

**Figure 10.** As in Fig. 9, but for subsets of events from those used in Fig. 9 that are (a) weakly associated with the MJO and (b) strongly associated with the MJO.

As noted earlier, such patterns of organized convection in the Tropics are often associated with the convectively active phases of the MJO – phases 3-6 from Wheeler and Hendon (2004). Since the MJO represents the dominant intraseasonal variation of tropical convection and since the MJO has been shown to project Kelvin wave-like temperature structures into the stratosphere (e.g. Kiladis et al., 2005), we extend this analysis of convection-momentum flux linkage by seeking to determine what impact the MJO has on the lower stratospheric momentum flux. We do so by partitioning our selected events into terciles based on the OLR MJO Index (OMI) values within ±5 days of the event date, the lower and upper thirds representing events that are weakly

and strongly associated with the MJO, respectively. Of the 29 weakly associated events, 14 do not have OMI values above 1 (active MJO) within $\pm$ 5 days. In contrast, 27 of the 29 strongly associated events have OMI values above 2 (strongly active MJO) over the same span. Furthermore, 23 of the 29 strongly associated events fall within the convectively active MJO phases 3-6. We are therefore confident that the OLR values used to identify the convective events that are also strongly associated with the MJO are being strongly affected by the active MJO.

Fig. 10 shows the composite momentum flux and zonal wind for both of these subsets of events. Both sets have significant positive anomalies in the 20-50 day range, in line with prior work showing that there is considerable Kelvin wave activity during MJO inactive periods (Virts 2014). However, the anomalies during events strongly associated with the MJO are much larger. The positive anomalies during these events begin prior to the event date, likely due to generation of flux by incipient convection upstream of the sounding site. As stated above, these results hold even if the whole of the Tropics is considered for determination of events so this is not a case of convection not yet entering the reference upstream area. It is likely that the convection that is forcing positive Kelvin wave momentum flux anomalies has not yet grown to cover a large enough area to exceed our threshold.

A second feature of Fig. 10 is the existence of significant positive anomalies above 20 km. These anomalies track downwards in time as the composite zonal wind increases, indicating that these flux anomalies are associated with the easterly phase of the QBO. Son et al. (2017) show that there is a discernable linkage between the phase of the QBO and the amplitude of the MJO, with stronger MJO in easterly QBO. Though these results may be a reflection of that linkage, our compositing does not allow for this kind of definitive attribution. It is not clear if these anomalies are resultant from the upstream convection and thus are tied to the MJO, if they are forced elsewhere and are able to propagate through the favorable mean state, or if they result from aliasing of the QBO into these composites. Methods to calculate anomalies of momentum flux with respect to the QBO would likely lead to a clearer composite signal, but doing so here would be dubious given that only four full QBO cycles are contained in these data.

Our method of compositing with this 11-year record of data is ultimately not sufficient to make definitive arguments about the impact of the MJO on lowermost stratospheric Kelvin wave momentum flux. Nevertheless, the above findings do suggest that the MJO is associated with anomalous increases in the flux, in at least the lowermost stratosphere.

## 7   Summary

We expand on prior methods for using high-resolution radiosonde observations to estimate upward fluxes of zonal momentum by Kelvin waves. Our methodology, in contrast to previous studies that used non-overlapping windows of data, makes use of short-time Fourier transform to generate daily time series of momentum fluxes that are useful for detailed analyses of intraseasonal and interannual variability. Unlike prior work using similar methods, we make use of relatively long-term sources of high-vertical resolution radiosonde data provided by the DOE ARM program to enable analysis of such variability. The qualitative nature of our derived time series is found to agree well with previous results – e.g. they show amplification during

both JFM and QBO easterly phases – and they qualitatively match prior estimates (cf. Ern and Preusse, 2009, and references therein) though with slightly different amplitudes.

Dependence of our results on vertical and temporal resolution is determined by reprocessing raw radiosonde data across different resolutions and comparing spatially and temporally overlapping points from our momentum flux calculation. Temporal resolution does not strongly affect the flux amplitudes for the tested time steps. In contrast, there is an approximately linear increase in the calculated flux amplitudes with increased vertical step, particularly beyond 500 m resolution (Fig. 2). The root of this relationship comes from our method of estimating the vertical wavelength $L_z$. For lower vertical resolution, less vertical structure is obtained and the calculation tends towards larger estimates of the vertical wavelength. These longer vertical wavelengths result in smaller values of wavenumbers $k$ and $m$, though the effect on $m$ is larger because it varies proportionally with $L_z$ (cf. Eq. (4)). From Eqs. (1) and (3), it is clear this will result in larger momentum fluxes for westerly background wind. From Fig. 6, the mean background wind in our data is westerly and thus our resolution tests show increases in momentum flux. In easterly background wind – where the Kelvin wave amplitudes are large – overestimation of the vertical length scales will result in smaller measured flux, underestimating the impact of the Kelvin waves.

Sensitivity to vertical stepping larger than 500 m highlights the need for continued collection of high vertical resolution observations in the tropical stratosphere. Both satellite observations and reanalysis reconstructions have much larger (order 1 km and larger) vertical stepping at these altitudes, so estimations derived from these sources alone may not fully capture the vertical structure. Perhaps more problematic is that the effect of coarse vertical resolution is not single-signed. Estimates will be too large in westerlies when the flux should be small and will be too small in easterlies when the flux should be large. Advances in remote sensing and computing capabilities will allow for smaller vertical stepping in both these platforms, helping to alleviate this sensitivity. However, there will still be a significant role for routine, high resolution radiosonde data in constraining satellite observations and in nudging data assimilation procedures for reanalyses.

By comparing calculated fluxes from highly-processed radiosonde data during the DYNAMO field campaign to calculated fluxes from synchronous raw radiosonde data at identical and nearby sounding stations, we find that our method is well-suited for application to these raw data, of which there is a considerably longer data record. Kelvin wave momentum fluxes are then calculated from an 11 year span (2003-2013) of quality radiosonde observations from this data record. The annual mean cycle of momentum fluxes shows an annual periodicity with a JFM maximum and a minimum six months later for July through September (Fig. 6). The QBO mean momentum fluxes are large during easterly phase and small during westerly phase (Fig. 7), as expected.

A composite analysis of events featuring broad deep convection shows that momentum fluxes are significantly increased by 30 days following the onset. Though the signal is only significant up to ~20 km, the amplitude of the flux increases by ~0.5-0.75 standard deviations in this region. By binning these events by OMI values and compositing, the data suggest that the MJO significantly increases the momentum flux relative to periods without an active MJO. Further study is necessary to more fully demonstrate the quantitative impact of the MJO.

Such studies should continue technique developments and data analyses that are necessary to further constrain the tropical stratospheric momentum flux budget. Techniques could be developed to incorporate simultaneous soundings from multiple sites

into a single calculation of momentum fluxes. The results derived here come from two radiosonde sites, but many additional sites with long data records are available. A careful reprocessing of these radiosonde data may allow for extending the data record with already available data.

Continued collection of high-quality radiosonde observations that probe the tropical stratosphere will also be vital for increasing the number of observed annual and QBO cycles. In addition to the role radiosondes have in forecasting (e.g. Cardinali and Healy, 2014) and climatological studies (e.g. Seidel et al., 2012), recently observed anomalous evolution of the QBO (Newman et al., 2016) further bears out the need for continued radiosonde observations. Additional observations in this region will allow for more robust attribution of wave sources – whether Kelvin, Rossby, or gravity waves – to changes in zonal wind structure in the tropical stratosphere.

*Competing interests.* The authors declare that they have no conflict of interest.

*Acknowledgements.* The authors would like to acknowledge Paul Ciesielski for his help in obtaining the L4 DYNAMO data and the raw radiosonde data. The analysis of Section 6 benefited from discussions with Zhen Zeng. We would also like to thank two anonymous reviewers who provided many helpful comments during the revision of this manuscript. This research was supported by Department of Energy Atmospheric System Research Grant DE-SC0008582.

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
