# Peer review of "Intraseasonal to interannual variability of Kelvin wave momentum fluxes as derived from high-resolution radiosonde data"

_Atmospheric Chemistry and Physics, 2016_

## Referee Comment (RC1) · Anonymous Referee #3 · 16 Jan 2017

Review comments for "intraseasonal to interannual variability of Kelvin wave momentum fluxes as derived from high-resolution radiosonde data" by Sjoberg et al.

This manuscript seeks to derive the Kelvin wave momentum flux (KWMF) vertical profile from a series of radiosonde observations. The methodology is first developed and tested about its robustness against high-vertical-resolution, high-temporal-resolution, intensive radiosonde obs. During the DYNAMO campaign, and then applied to two low-vertical-resolution, low-temporal-resolution, long data record ARM radiosonde datasets for the sake of studying the intraseasonal and interannual variabilities. The authors focused on studying the QBO and MJO's impacts on the KWMF. For QBO, the finding agrees qualitatively with many previous studies that KWMF plays a major role in the descent of the QBO westerly phase; for MJO, the authors found that there is a nontrivial (but statistically insignificant) increase in KWMF when MJO is in its active phase versus the inactive phase.

While the value of using high-resolution radiosonde data to study (and to separate out the Kelvin waves in the first sense) the KWMF and its interaction with other tropical variabilities is highly appreciated, the originality of this work is relatively weak. The methodology is refined from Sato and Dunkerton (1997) and Holton et al. (2001), and the major findings are mostly "qualitatively" agree with previous other findings. Of course the datasets employed here are unique. I think the novelest finding I appreciated the most as well is that the sensitivity of the magnitude of KWMF to the vertical and temporal intervals are investigated, which could greatly help us understand the discrepancies among the values calculated using different obs. or modeling techniques. The authors should pay extra effort in the revised manuscript to point out the uniqueness and importance of this work.

Overall the writing is OK, and the logic flow is natural. I found tiny inconsistencies from places to places that may confuse the readers though.

Now, since this review is for publication pending to ACPD, I'd point out some re-arranging suggestions first before raising my scientific concerns. I have no problem of publishing it on ACPD first after revising some of the awkward logics pointed below, and the authors can then prepare and think more about my further scientific comments.

**Re-arrange some of the sentences/figures, and correct grammar errors.**
1) Fig. 1: please consider at a panel of background wind contours so we know what the background wind looks like, and more importantly, we can see whether its varying slowly vertically.
2) Fig. 2: add the errorbar for each line for each resolution you picked to construct the lines. Since I'm not clear how many soundings were used, if only a few, you need to explain how robust Fig. 2's results are to a large sample; if a lot of the soundings were used, you can comfortably plot the errorbars out.
3) Fig. 3: add the explanation of the bold grey line (zero wind line) or add the label in the figure.
4) P13, L4: it would be much straightforward if you can show a scatterplot of your comparison between the two datasets.

5) Fig. 7: since you stated that 8-20day Kelvin waves are representative of the total KWMF features of 5-20 day Kelvin waves, and there are a lot of missing data for 5-8 day Kelvin waves using your technique, why not revised Fig. 3, 4, 5, 6 with the 8-20 day Kelvin waves? I think it's very important to keep consistency throughout the paper of the variables you present. Otherwise, you don't know whether the differences are caused by other mechanisms or simply by the inconsistency. Move your explanation of P15, L11-15 to the second paragraph of Section 4.

6) It's very awkward to further extend your discussion about MJO's impact on KWMF in the conclusion section. Why not move this discussion to a subsection of Section 5 (also change the title of Section 5)?

7) You mentioned you used two indices to indicate the phase of MJO: RMM and OMI. Firstly, you need to clarify which datasets are used to construct these two indices; secondly, you don't even used RMM throughout the paper, if I didn't read too fast to miss that point. Please point the sentences about RMM out in the MJO section.

8) Fig. 9: I don't understand the meaning of the x-axis of Fig. 9b. Can't imaging the errorbar could be uniform throughout the layer. Can you add your spread (2*sigma or 3*sigma) to Fig. 9a for both solid and dashed lines (since you have a lot of sounding profiles to composite each of the line), so it would be much more straightforward to check whether they are statistically different.

9) minor grammar errors:
P1, L9: reveals -> reveal that
L10: add "in" before "boreal"
L11: add "the downward propagation of the" before "easterly".
L11: add "the" before "Madden".
L12: remove "the" before "MJO". Add "the" before "lowermost".
L16: remove "to".

P3, L6: add "the" before "reanalyses".
L22: add "the" before "collection".
L29: Since balloon ascent rate is ~ 5m/s, 2-sec resolution means the vertical resolution should be around 10 m? Why you indicated later that it ranges between 50 and 2000 km?

P4, L22: In the description of methodology section, you mentioned that the rationale of choosing the vertical resolution of 250m will be discussed in Section 3. Where in Section 3 did you discuss your motivation? I'm sorry I couldn't find it. Also, just as I said in the last comment, why not use even higher vert. resolution, or at least test the sensitivity to a smaller vertical interval of KWMF?

P9, L7: "the blue curve": which curve? Fig. 2 is black and white.

P11, L1: "removed" -> "away".

P18, L2: Since "OMI" has been mentioned long ago, it's better to refresh the readers' memory of re-spell out what the word stands for and the reference.
L1: move L5's explanation of the dashed line here and combine it with the first sentence of this paragraph.
L17: "Fig. 6": can you double check if you get the figure number correct?

P19, L29: add your funding source if there's any.

**Now, major comments (can be answered after the publication on ACPD):**
1) How good is your assumption that the vertical wavenumber is constant for a given window of data? My understanding is that you still estimate the vertical wavenumber for each period of Kelvin wave (5-20 days) separately, is that correct? How to justify the impact if it is not the case? Can you assess how many cases in terms of percentage of total that violates the slow-varying-zonal-wind rule (is this the WKB assumption by the way)?

2) Please add a sentence or two to clarify that easterward propagating gravity waves (GWs) would not be included in your KWMF calculation, as you only constrain your horizontal wavelength to be > 100 km, fairly fall in the spectrum of internal and inertial GWs.

3) Fig.2: it would be the best to add a panel showing how your vertical wavelength (Lz) change with decreasing the vertical resolution for, e.g., 5-day, 10-day, 15-day, and 20-day waves.

4) P13, L2: I strongly suggest you to elaborate the reason to explain the discrepancies among different datasets here more thoroughly, e.g., SABER retrieved temperature profiles or ERAi might have too coarse vertical resolution, etc. Then briefly summarize this point in the conclusion section.

5) P15, L16: I don't quite understand. KWMF plays a critical role in the descending of the QBO westerly phase, which shows a discernable enhancement along the zero-wind-line, as also shown in Fig. 4 and Fig. 5. Then, when you do the composite, it seems to me that the KWMF enhancement actually occurs when the QBO easterly starts to weaken. Why?

6) P17, L28 and onward about the MJO discussion: firstly, you need to give a reference or two suggesting that MJO likely impacts the KWMF. As you later on stated that some of the previous studies also found that Kelvin waves were also released when MJO was in the inactive phase: then why conduct such an investigation?

If you'd like to study whether convective activities are tied to KWMF strength, simply use the daily OLR index for a given grid box around the sounding site, and set up a threshold to separate active and inactive convective days to composite the KWMF.

7) Like I said in the beginning, add some sentences or paragraphs highlight the uniqueness and novelty of your work.

---

## Referee Comment (RC2) · Anonymous Referee #1 · 11 Feb 2017

This paper describes a method to estimate time series of Kelvin wave momentum flux from radiosonde data, and by applying the method, seasonal and QBO-related variations of the flux are obtained using 11-year sonde data. In addition, the sensitivity of the estimates to the vertical and temporal resolutions is assessed. The flow of the text is logically natural, and the figures attached are high quality. The method used seems proper and advantageous to obtain continuous time series of flux for target period bands, although it also has a limitation for easterly background winds (see the specific comments below). I recommend this paper for publication after revisions regarding the following comments.

[ specific comments ]

1. In section 2.2, the authors describe in detail the estimation method which is extended from the method used in previous studies. I suggest clarifying the difference, extension, or improvement from the previous method. For example, in P5 L28–31, the authors state that the results from their method are similar to those from previous studies in terms of overall range of vertical wavelengths, confirming the fidelity of the method. However, it is not clearly stated what the improvement/advantage of the present method is. Clarifying this in section 2.2 and/or in conclusion section could help readers and strengthen the paper.

2. P4 L20–23: To demonstrate resolution effects more completely, the sensitivity of estimates not only to output resolutions ($\Delta t_{win}$ and $\Delta z_{win}$) but also to raw data resolutions (i.e., vertical/temporal stepping of raw data before interpolation procedure) could be investigated. For example, from a 50-m resolution profile, one could make a 300-m resolution profile by picking one data point every six points. Interpolation using the original 50-m data and that using the sub-sampled 300-m data can result in different estimates of parameters even for the same $\Delta z_{win}$ value.

3. P6 L6: "strong easterlies often result in negative k." : This can be in part due to the restriction of ground-based frequency to be positive. In principle, the spectral transform in time just gives the absolute value of the frequency, so that we still have freedom to determine its sign, while the intrinsic frequency (and k) is fixed to be positive. What will happen in the results if negative ground-based frequencies are allowed in the strong easterly regions ?

4. Figs. 4 and 5: Too many regions are filled by missing for the easterly wind where the Kelvin wave flux is actually maximal (e.g., Ern and Preusse, 2009). Also, the regions

of large momentum flux in the westerly shear layer, which are important for the QBO to descend, are very close to the missing regions below. Therefore, the large Kelvin wave flux in such strong easterly regions could be of interest. It would be very nice if the authors explore ways to estimate the momentum flux in such easterly regions, as much as they can.

5. I feel that the grammar used is not perfect. The judgment for English editing will be left to the authors and other reviewers.

[ minor comments ]

P1 L4-5: "Estimates . . . larger." : Readers could read this as the authors themselves also estimated the momentum flux from satellite and reanalysis data. I suggest deleting this sentence.

P2 L5: "identical" → "opposite" ? Please check this and make it consistent with the descriptions in this paragraph.

P2 L8–17: Some phrases are repetitive within this paragraph. Please reorganize this paragraph.

P2 L19: "vertical momentum" → "zonal momentum"

P3 L9: I suggest including ", variability, " between "climatologies" and "vertical ...", considering the title of this paper.

P3 L16: What do the "two climatologies" mean ?

P3 L29: What is an approximated vertical step corresponding to the 2 seconds, considering lifting speed of the balloon ?

[Figure]

P4 L13: "linearly interpolated in height and spline interpolated in time": The linear interpolation also is one of the spline interpolations. Please include the order of the spline interpolation in time used here (e.g., cubic spline).

P4 L17: "linear and spline" → "orders of" / "changes to" → "changes in"

P4 L18: "point. An exception to this is if the time" → "point, unless the time"

P4 footnote: "too short" → "too long" ? Based on my experience, the scale height in the tropical lower stratosphere is about 6 km or even shorter.

P5 L6: "but that variations in the stratification ... $L_z$." : Where (and how) is this assumption used in your method ?

P5 L25: "temperature leads zonal wind": What is the criterion for this lead/lag relation ? e.g., phase difference of 45–135°, or 0–180° ? It is better to include this information in the text, considering that the determination of lead/lag relation between two variables is ambiguous as the phase difference becomes close to 0 or 180°.

P5 L27: Please include the minus sign in front of the "$2\pi$", as the authors defined m to be negative (P5 L5).

Fig. 1 caption: Please include "40-day mean" in (c) in front of "vertical quadrature spectrum". In addition, I suggest changing "filtering window" to "period" (L5; L6; L8) in order to clarify its meaning.

P8 L6: "as expected ... (a)" : Zonal wavelengths cannot be expected from visual inspection of (a) in which the time–height cross section is shown.

Fig. 2: The right axes are not linear while the left axes are linear. I have thought that the percent difference is defined as $(M - M_0)/M_0$ where $M_0$ is the momentum flux estimated with the reference (250-m and 24-hour) resolutions. If it is right, the percent difference and M have linear relationship.

Fig. 2 caption: "time mean momentum fluxes from ..." : Based on the text, it is more

precise to describe this as "momentum flux, estimated using time-mean parameters, from ..."

P9 L7–10: As already pointed out by the technical review of the manuscript, there is no curve in the figure that the authors describe in these sentences. The dashed curve, which is referred to by these sentences, is totally different one, as mentioned in the figure caption and on P9 L5.

P9 L18: "enhanced" : What does this mean ?

P10 L5: "full zonal mean" : I do not agree to use the term "zonal mean momentum flux" for the flux estimated using one-site data, as here. The temporal mean could approximate the zonal mean for zonal wind or temperature in the stratosphere, as mentioned by the authors, but it could not approximate the zonal mean of anomaly flux in general. Please consider revising this, as well as in P17 L13–14.

Fig. 6: Could you explain why the parameters in (b)–(d) are weighted by period ? (i.e., reason why the parameters with longer periods are more highlighted)

P14 L1: "the westerly QBO phase persists longer in the lower stratosphere" : This could be partly due to the missing when the wind is easterly. Or, is the zonal mean zonal wind here composited regardless of the missing for momentum flux estimates?

P15 L12: For given zonal mean N and U, the sign of k (i.e., missing or not) depends only on the magnitude of m by Eq. (4), as the authors fix $\omega$ to be positive. Thus, the numerous missing for the 5–8 day period bands may imply that for these short periods the vertical wavelengths are shorter than those for 8–20 day waves. Is this true overall ? It seems to be not the case for the example in Fig. 1.

Fig. 7: While the climatological momentum flux is much larger below 20 km than above as shown in Fig. 6a, the flux below 20 km shown in Fig. 7 seems not that large compared to above, even when averaged over the QBO phases. Does this imply that a large portion of the flux below 20 km shown in Fig. 6a comes from the 5–8 day waves

that are excluded in Fig. 7 ?

P15 L13: "The same structure" : same as what ?

P15 L18–P16 L1: "signals of downward descending fluxes" → "descending signals of the flux"

Fig. 8: Based on the positions of number of days indicated in this figure and based on the shape of the contours in Fig. 7, I assume that the QBO phase bins are centered at 0, 0.25, 0.5, 0.75, and so on. However, the histogram in Fig. 8 is centered at 0.125, 0.375, and so on. Please correct the figure.

P16 L13: "for linear . . . resolution" → "with increased vertical step"

P16 L17: Please insert "for westerly background wind" after "results in larger momentum fluxes", because there is the inverse relationship for easterly cases (Eq. (4)).

P17 L24: "planetary-scale, zonal mean momentum fluxes" → "planetary-scale wave momentum fluxes"

P18 L5: "MJO is" → something like "active-MJO mean is"

P19 L7: "As shown by Fig. 9" → "As mentioned" (It was "not shown").

[ typos / technical corrections ]

P1 L1: "estimates . . . remains" : plural/singular

P1 L8: "the" → "a"

P1 L9: "ARM" : The full name of the ARM is not introduced. Also, it is not clear in this sentence whether the ARM site data is by the DOE or DYNAMO.

P1 L9: Delete "available".

P1 L11: Delete "(QBO)".

P1 L24: "Qausi-Biennial" : change this to small letters, as in the abstract.

P2 L1: "is an . . . connection" → "are . . . connections"

P2 L11 and L12: "waves with phase speed in the direction of mean flow" : change "speed" to "velocity" (because speed cannot describe the direction), or change the phrase to "waves propagating to the direction of ..."

P3 L5: "show" → "showed"

P3 L22 and P4 L5: Delete "Madden Julian Oscillation", while remaining "MJO" (as already introduced in P1 L19).

P4 L7: "empirical"

P4 L14: "has"

P4 L28: The meaning of the symbol "T" is not introduced.

P4 L29: "density" → "density ($\rho$)" / "from which" → "of which"

P5 L29: "nearly the same as" → "close to" (One may not agree that 4.5 km and 4.0 km are nearly the same.)

P8 L19: "coexistence . . . occur" : singular/plural

P8 L21: "are increasingly" → "increase"

Fig. 2 caption, L3: "250 m" → "24 hour"

P11 L5: "Nauru"

P13 L10: "1" → "(1)"

Eq. (5): Please insert "$1/\pi$" in front of "$\tan^{-1}$".

P16 L7: "nature . . . are" : singular/plural

Fig. 9 caption: "16" → "15" (based on the text, P18 L4; P19 L9).

P18 L14: "showed"

---

## Author Comment (AC1) · 16 May 2017

"Intraseasonal to interannual variability of Kelvin wave momentum fluxes as derived from high-resolution radiosonde data"
ACP-2016-1088

We want to thank both the reviewers for their helpful comments, suggestions, and corrections to the manuscript. We feel that the draft is greatly improved as a result of all of these and hope that it more thoroughly presents our analysis and more precisely explains the results. Due to the great detail of the reviewer responses, the manuscript has undergone a major revision. As such, we would like to first describe and explain the large changes we have made.

Thanks to comment 3 from Reviewer #1, the number of valid data points have been increased. Selection of the sign of omega allows us ensure that k remains positive. None of the results – most importantly, the raw time series – showed that making this change produces unphysical or unexpected results, except in regions where the horizontal wavelength is less than 100 km. In these regions, there are likely spurious oscillations of the sign of our estimated momentum flux. Furthermore, it is not clear that the waves in these regions are truly Kelvin waves given that at 100 km length scales, Kelvin waves are indistinguishable from gravity waves. As a result, we increased the length constraint to 500 km and this removed most of the oscillatory values, while leaving more resolved points within strong easterlies.

Naturally, this methodological change alters many details of our results. For instance, the composite structures change considerably. In the case of the QBO composite, these changes result in a composite structure that even more closely matches expectations and removes the problem that including wave periods between 5-8 days produces missing values in the composite. This latter improvement addresses a comment by Reviewer #2 about excluding those periods from the rest of our analysis.

In the case of the annual cycle composite, the changes are perhaps more noticable, particularly those for zonal wind in the top left panel. We note that, for the annual mean composite, we only include points in the composite where the momentum flux is not a missing value. This is true for each field, not just momentum flux itself. In the previous submission, because many missing values were located in the easterlies, the composite zonal wind structure came from points that were primarily in QBO westerlies or during transition periods. Now that we include additional data points from the QBO easterlies, this brings about the differences in the composite structure.

The suggestion by Reviewer #2 that, before attempting to make the link between the MJO and momentum flux, we first analyze the expected relationship between convection and Kelvin waves also produced a substantive change to the manuscript. The details of this are now included in the Discussion section – separated from the Summary as per a comment by Reviewer #2 – but we can give an overview here.

We determined an appropriate definition for the occurrence of a convective 'event' upstream of the sounding sites and performed composite analysis about these events. This shows that organized convective events precede a positive signal in lower stratospheric momentum flux, as expected. By then comparing those convective events with strong and weak associations to the MJO, we find that the strongly associated events have considerably more flux. This provides a more definite suggestion that the MJO influences stratospheric momentum flux. We believe that a key shortcoming to our previous analysis of this was that we did not require a convective signal (i.e., a forcing mechanism) to be present near the sounding site.

We have addressed your individual comments below in blue.

Reviewer #1

1. In section 2.2, the authors describe in detail the estimation method which is extended from the method used in previous studies. I suggest clarifying the difference, extension, or improvement from the previous method. For example, in P5 L28–31, the authors state that the results from their method are similar to those from previous studies in terms of overall range of vertical wavelengths, confirming the fidelity of the method. However, it is not clearly stated what the improvement/advantage of the present method is. Clarifying this in section 2.2 and/or in conclusion section could help readers and strengthen the paper.

R: Thank you for this comment. We were not clear enough about what is different from prior studies. Our primary difference is just that we use overlapping windows, here overlapping by all but 1 day from the ones prior to or following a given window. This technique of short-time Fourier transform allows for greater temporal resolution of which previous studies, to our knowledge, have not taken advantage. We have added a few sentences to the end of section 2.2 and to the summary in order to clarify this to the reader.

2. P4 L20–23: To demonstrate resolution effects more completely, the sensitivity of estimates not only to output resolutions ($\Delta t_{win}$ and $\Delta z_{win}$) but also to raw data resolutions (i.e., vertical/temporal stepping of raw data before interpolation procedure) could be investigated. For example, from a 50-m resolution profile, one could make a 300-m resolution profile by picking one data point every six points. Interpolation using the original 50-m data and that using the sub-sampled 300-m data can result in different estimates of parameters even for the same $\Delta z_{win}$ value.

R: we have performed this analysis and not found the results to be different. As an example, the figures below show the comparison of our (left) original 250 m data and 250 m data computed using input data with at least 300 m stepping and (right) the same but for 500 m

resolution and stepping. These two data sets have a linear correlation coefficient of 0.945 and 0.881, respectively. This seems to indicate that there is not high sensitivity to the resolution of the input data, and that changes to the resolution of the input data do not systematically alter the resulting momentum flux estimates.

[Figure]

3. P6 L6: "strong easterlies often result in negative k." : This can be in part due to the restriction of ground-based frequency to be positive. In principle, the spectral transform in time just gives the absolute value of the frequency, so that we still have freedom to determine its sign, while the intrinsic frequency (and k) is fixed to be positive. What will happen in the results if negative ground-based frequencies are allowed in the strong easterly regions?

R: Thank you for this insight. We overlooked the possibility of applying this freedom afforded by spectral symmetry. We have discussed this change in our summary above, and made appropriate changes to the text.

4. Figs. 4 and 5: Too many regions are filled by missing for the easterly wind where the Kelvin wave flux is actually maximal (e.g., Ern and Preusse, 2009). Also, the regions of large momentum flux in the westerly shear layer, which are important for the QBO to descend, are very close to the missing regions below. Therefore, the large Kelvin wave flux in such strong easterly regions could be of interest. It would be very nice if the authors explore ways to estimate the momentum flux in such easterly regions, as much as they can.

R: We agree that the fluxes in these regions are of interest. Our original response would have been to discuss that inclusion of additional radiosonde sites in a merged analysis would allow for independent estimation of the zonal wavelengths. This independent estimation could reduce the frequency of negative values of k in regions of strong easterlies/westerly shear, allowing our method to then estimate the momentum flux there. While we think that this is a

worthwhile procedure for future analysis, addressing your previous comment helped to greatly reduce the number of missing values.

5. I feel that the grammar used is not perfect. The judgment for English editing will be left to the authors and other reviewers.

R: we have performed a thorough reading of the text, paying particular attention to improving grammar and sentence structures.

P1 L4-5: "Estimates . . . larger." : Readers could read this as the authors themselves also estimated the momentum flux from satellite and reanalysis data. I suggest deleting this sentence.

R: deleted.

P2 L5: "identical" → "opposite" ? Please check this and make it consistent with the descriptions in this paragraph.

R: this was overlooked, but is now changed to be correct.

P2 L8–17: Some phrases are repetitive within this paragraph. Please reorganize this paragraph.

R: we have edited the text somewhat in this paragraph, but retained most of it because we feel that a thorough explanation is worthwhile here.

P2 L19: "vertical momentum" → "zonal momentum"

R: changed.

P3 L9: I suggest including ", variability, " between "climatologies" and "vertical ...", considering the title of this paper.

R: this is a very welcome suggestion. Changed.

P3 L16: What do the "two climatologies" mean?

R: we mean the climatological annual cycle and the QBO mean cycle. We have made this more clear at this point, and changed the title of section 5 (to "Annual cycle and the QBO") for more clarity.

P3 L29: What is an approximated vertical step corresponding to the 2 seconds, considering lifting speed of the balloon?

R: on average, about 10 m. We have put this approximate value in the text.

P4 L13: "linearly interpolated in height and spline interpolated in time": The linear interpolation also is one of the spline interpolations. Please include the order of the spline interpolation in time used here (e.g., cubic spline).

R: thank you for this clarification. We have corrected the text.

P4 L17: "linear and spline" → "orders of" / "changes to" → "changes in"

R: changed.

P4 L18: "point. An exception to this is if the time" → "point, unless the time"

R: we feel that this would create a run-on sentence. We have altered the sentence in a way that we hope is clearer.

P4 footnote: "too short" → "too long" ? Based on my experience, the scale height in the tropical lower stratosphere is about 6 km or even shorter.

R: we have made this change.

P5 L6: "but that variations in the stratification ... $L_z$." : Where (and how) is this assumption used in your method ?

R: this is the WKBJ assumption. It was used in the derivation of Eqs. (1)-(4), principally through the wavelike approximations made for each geophysical field. We added more discussion about the use of this assumption to the text.

P5 L25: "temperature leads zonal wind": What is the criterion for this lead/lag relation ? e.g., phase difference of 45–135°, or 0–180° ? It is better to include this information in the text, considering that the determination of lead/lag relation between two variables is ambiguous as the phase difference becomes close to 0 or 180°.

R: we have previously tested the difference between these two criteria and found the results to not be significantly different. However, you make an excellent point that we should err on the side of certainty in this lead/lag relation. We have updated our method and included this information in the text.

P5 L27: Please include the minus sign in front of the "$2\pi$", as the authors defined m to be negative (P5 L5).

R: done.

Fig. 1 caption: Please include "40-day mean" in (c) in front of "vertical quadrature spectrum". In addition, I suggest changing "filtering window" to "period" (L5; L6; L8) in order to clarify its meaning.

R: we have made these changes.

P8 L6: "as expected … (a)" : Zonal wavelengths cannot be expected from visual inspection of (a) in which the time–height cross section is shown.

R: indeed. We have made this change.

Fig. 2: The right axes are not linear while the left axes are linear. I have thought that the percent difference is defined as $(M - M_0)/M_0$ where $M_0$ is the momentum flux estimated with the reference (250-m and 24-hour) resolutions. If it is right, the percent difference and M have linear relationship.

R: Percent difference, to our knowledge, is defined as (M-M0)/ ( (M+M0)/2 ), while percent error is defined in the way you stated.

Fig. 2 caption: "time mean momentum fluxes from …" : Based on the text, it is more precise to describe this as "momentum flux, estimated using time-mean parameters, from …"

R: we agree with this correction, but the method of calculating the momentum flux for this experiment has changed to instead use the daily values of the parameters and then calculate the time mean.

P9 L7–10: As already pointed out by the technical review of the manuscript, there is no curve in the figure that the authors describe in these sentences. The dashed curve, which is referred to by these sentences, is totally different one, as mentioned in the figure caption and on P9 L5.

R: thank you for noticing this. It has been removed.

P9 L18: "enhanced" : What does this mean ?

R: it is more clear to say "positive" so the text now uses this instead of "enhanced."

P10 L5: "full zonal mean" : I do not agree to use the term "zonal mean momentum flux" for the flux estimated using one-site data, as here. The temporal mean could approximate the zonal mean for zonal wind or temperature in the stratosphere, as mentioned by the authors, but it could not approximate the zonal mean of anomaly flux in general. Please consider revising this, as well as in P17 L13–14.

R: while the derivation of Eqs. (1)-(4) are for the zonal mean momentum flux, we acknowledge that it may be misleading to call our single-site estimates of the flux a true zonal mean. It's not clear (and we have not shown) how these long-term estimates vary along longitude, other than for an additional site located 20 degrees downstream. We have removed the wording as you suggested.

Fig. 6: Could you explain why the parameters in (b)–(d) are weighted by period ? (i.e., reason why the parameters with longer periods are more highlighted)

R: considering it now, this does not seem to be a sensible thing to do. Since panels (c) and (d) are not much different for a simple mean over all wave periods when compared to the period-weighted mean, we elect to do the former in the manuscript.

P14 L1: "the westerly QBO phase persists longer in the lower stratosphere" : This could be partly due to the missing when the wind is easterly. Or, is the zonal mean zonal wind here composited regardless of the missing for momentum flux estimates?

R: this is precisely the reason. Now, with more included points going into the annual mean of the zonal wind (because more momentum flux points are not missing), there is a much larger signal of the QBO easterlies. While it's still true that the westerly phase persists longer over our data record (see Figs. 3 and 4), the easterly winds are stronger. The text now reflects this.

P15 L12: For given zonal mean N and U, the sign of k (i.e., missing or not) depends only on the magnitude of m by Eq. (4), as the authors fix $\omega$ to be positive. Thus, the numerous missing for the 5–8 day period bands may imply that for these short periods the vertical wavelengths are shorter than those for 8–20 day waves. Is this true overall? It seems to be not the case for the example in Fig. 1.

R: this is not true overall. There is no forced or implied dependence of vertical wavelengths on wave periods, though this tends to be generally true as can be seen in Fig. 2b. Note though that, in the mean, the 20 day waves have shorter scales than 13.3 day waves.

Fig. 7: While the climatological momentum flux is much larger below 20 km than above as shown in Fig. 6a, the flux below 20 km shown in Fig. 7 seems not that large compared to above, even when averaged over the QBO phases. Does this imply that a large portion of the flux below 20 km shown in Fig. 6a comes from the 5–8 day waves that are excluded in Fig. 7?

R: while this particular issue has been removed from the manuscript, we can inform you that, in the annual mean, these waves periods account for approximately 45-55% of the total momentum flux between 18 and 20 km.

P15 L13: "The same structure" : same as what ?

R: removed, so no longer applicable. Referred to the 5-20 day and the 8-20 day QBO composite having the same structure.

P15 L18–P16 L1: "signals of downward descending fluxes" → "descending signals of the flux"

R: changed.

Fig. 8: Based on the positions of number of days indicated in this figure and based on the shape of the contours in Fig. 7, I assume that the QBO phase bins are centered at $0, 0.25, 0.5, 0.75$, and so on. However, the histogram in Fig. 8 is centered at $0.125, 0.375$, and so on. Please correct the figure.

R: Fig. 7 should be centered at the half steps 0.124, 0.375, etc. This is now corrected.

P16 L13: "for linear . . . resolution" → "with increased vertical step"

R: changed.

P16 L17: Please insert "for westerly background wind" after "results in larger momentum fluxes", because there is the inverse relationship for easterly cases (Eq. (4)).

R: we agree that this clarification should be included, and have amended the text. We also briefly discuss that the opposite is true, as this is important for the relation between Kelvin waves and the QBO.

P17 L24: "planetary-scale, zonal mean momentum fluxes" → "planetary-scale wave momentum fluxes"

R: amended.

P18 L5: "MJO is" → something like "active-MJO mean is"

R: no longer included.

P19 L7: "As shown by Fig. 9" → "As mentioned" (It was "not shown").

R: no longer included.

Reviewer #2

1a) Fig. 1: please consider at a panel of background wind contours so we know what the background wind looks like, and more importantly, we can see whether its varying slowly vertically.

R: Ensuring that WKBJ holds is an important aspect of this work, else our wave-like assumptions do not hold. In this case, the winds are slowly varying during this time span, as can be seen in Fig. 3. Because the winds are displayed there, we opt not to include an additional panel, but have added a note to the reader that they may see the wind over this data window in Fig. 3.

2a) Fig. 2: add the errorbar for each line for each resolution you picked to construct the lines.

Since I'm not clear how many soundings were used, if only a few, you need to explain how robust Fig. 2's results are to a large sample; if a lot of the soundings were used, you can comfortably plot the errorbars out.

R: we first took the time mean of each of the input fields to Eqs. (1) and (3). Doing so allows for very quick calculation of the momentum fluxes for our experiments holding certain input fields constant (e.g. vertical wavelength in Fig. 2). Since we first took the time mean, this obviously did not allow for any type of error analysis. We reformulated this portion of the analysis to instead calculate the momentum flux first, allowing us to then calculate standard errors. Note that this methodological change, along with changes arising from allowance of negative k values, results in changes to the overall amplitude of our results. From the standard error bars, it should be more apparent that the differences between 250 m and 2000 m resolution are significant. We have added text describing this.

3a) Fig. 3: add the explanation of the bold grey line (zero wind line) or add the label in the figure.

R: thank you for noticing this. We have corrected it here and elsewhere.

4a) P13, L4: it would be much straightforward if you can show a scatterplot of your comparison between the two datasets.

R: we don't feel that this substantively adds to the paper enough to justify a figure in the manuscript, but have made clear that the linear correlations between all points 18-25 km are high (0.94). We include the scatterplot below for your consideration, however (units mPa; gray is 1-to-1; black is linear fit).

[Figure]

Manus momentum flux

5a) Fig. 7: since you stated that 8-20day Kelvin waves are representative of the total KWMF features of 5-20 day Kelvin waves, and there are a lot of missing data for 5-8 day Kelvin waves using your technique, why not revised Fig. 3, 4, 5, 6 with the 8-20 day Kelvin waves? I think it's very important to keep consistency throughout the paper of the variables you present. Otherwise, you don't know whether the differences are caused by other mechanisms or simply by the inconsistency. Move your explanation of P15, L11-15 to the second paragraph of Section 4.

R: We agree that consistency is vital for proper comparison. Due to changes to method – allowing of negative wave periods in the calculation of k – this is no longer necessary. Kelvin waves with periods between 5-8 days now have a sufficiently large number of valid points to not result in missing values in the QBO composite.

6a) It's very awkward to further extend your discussion about MJO's impact on KWMF in the conclusion section. Why not move this discussion to a subsection of Section 5 (also change the title of Section 5)?

R: with the changes to this analysis, we definitely agree. We have broken out the discussion section (focusing on the relation between convection/MJO and momentum flux) from the summary.

7a) You mentioned you used two indices to indicate the phase of MJO: RMM and OMI. Firstly, you need to clarify which datasets are used to construct these two indices; secondly, you don't even used RMM throughout the paper, if I didn't read too fast to miss that point. Please point the

sentences about RMM out in the MJO section.

R: we did mention that results are not qualitatively different, but this is not really sufficient for including it here. We have removed mentions of the RMM. We also provided more details about the OMI data we use.

8a) Fig. 9: I don't understand the meaning of the x-axis of Fig. 9b. Can't imaging the errorbar could be uniform throughout the layer. Can you add your spread (2*sigma or 3*sigma) to Fig. 9a for both solid and dashed lines (since you have a lot of sounding profiles to composite each of the line), so it would be much more straightforward to check whether they are statistically different.

R: this figure has been removed.

1) How good is your assumption that the vertical wavenumber is constant for a given window of data? My understanding is that you still estimate the vertical wavenumber for each period of Kelvin wave (5-20 days) separately, is that correct? How to justify the impact if it is not the case? Can you assess how many cases in terms of percentage of total that violates the slow-varying-zonal-wind rule (is this the WKB assumption by the way)?

R: the WKBJ approach has been used to derive the equations (1)-(4). As is typically the case, real data does not fit as well as would be liked to the approximations made by this theory. Nevertheless, ensuring that it holds to first order can still yield insight we would not otherwise be able to get (see Andrews et al. 1987, sec. 4.7.4). This condition of slowly-varying zonal wind is one of the assumptions used by the WKBJ approach.

We can estimate how often this assumption is satisfied by first finding the appropriate scales of zonal wind variations. To find the length (time) scale, we take the 1-sigma value of zonal wind (15 m/s) and divide by the local vertical (temporal) derivative. To first order, "slowly varying" is satisfied where these scales are larger than the longest vertical or time scales we consider (15 km or 20 days). These conditions are satisfied in time and space for 99.9% and 85.5% of points, respectively.

As discussed in the aforementioned section by Andrews et al., WKBJ theory has been successfully applied to problems which have not entirely satisfied the assumptions. And, since we find good agreement between our and other authors' work, we feel justified in using these methods despite their shortcomings. If there were glaring unphysical results in our KWMF data, we would be much more cautious about interpretation and utilization of our results. Most features are quite reasonable, however.

Further, these shortcomings motivated us to not focus on specific events, but to instead perform more climatological/composite analysis. By performing our analyses only over a large number of events/cycles, unphysical estimates from where the applied model is invalid should get wiped out. While this is not true in the case of the QBO composite (with only 4 full events entering into it), the high number of points averaged in each bin raises our confidence in it.

We have added additional comments about the usage of WKBJ and our confidence in the results.

2) Please add a sentence or two to clarify that eastward propagating gravity waves (GWs) would not be included in your KWMF calculation, as you only constrain your horizontal wavelength to be > 100 km, fairly fall in the spectrum of internal and inertial GWs.

R: we have attempted to make this clearer where we describe the minimum required horizontal length scale (now taken to be 500 km).

3) Fig.2: it would be the best to add a panel showing how your vertical wavelength (Lz) change with decreasing the vertical resolution for, e.g., 5-day, 10-day, 15-day, and 20-day waves.

R: we agree and have added this panel to Fig. 2.

4) P13, L2: I strongly suggest you to elaborate the reason to explain the discrepancies among different datasets here more thoroughly, e.g., SABER retrieved temperature profiles or ERAi might have too coarse vertical resolution, etc. Then briefly summarize this point in the conclusion section.

R: since the background state for these values is easterlies, our resolution tests would suggest that larger vertical length scales should result in underestimation of the flux. We have added additional text about this wind state-resolution relationship in Section 3. Thus, while we acknowledge that coarse vertical resolution in these data will result in errors, it is not clear that Section 4 is the appropriate place to discuss this. Instead, we bring up the comparison to show that we are confident in our estimates. We have included additional text in the Summary giving further elaboration to these ideas, however.

5) P15, L16: I don't quite understand. KWMF plays a critical role in the descending of the QBO westerly phase, which shows a discernable enhancement along the zero-wind-line, as also shown

in Fig. 4 and Fig. 5. Then, when you do the composite, it seems to me that the KWMF enhancement actually occurs when the QBO easterly starts to weaken. Why?

R: in the previous versions of Figs. 4 and 5, there were too many missing points in the core of the easterlies to notice how large the fluxes were. This has been corrected for this new submission. You are correct that that the KWMF is critical in the transition, but the acceleration from the flux is strongest where (the negative of) the vertical gradient is largest. In Fig. 7, this occurs where along the regions of large vertical gradients in zonal wind. The text as written was not clear about this, so we have edited it.

6) P17, L28 and onward about the MJO discussion: firstly, you need to give a reference or two suggesting that MJO likely impacts the KWMF. As you later on stated that some of the previous studies also found that Kelvin waves were also released when MJO was in the inactive phase: then why conduct such an investigation?

If you'd like to study whether convective activities are tied to KWMF strength, simply use the daily OLR index for a given grid box around the sounding site, and set up a threshold to separate active and inactive convective days to composite the KWMF.

R: thank you for this suggestion. It seems obvious now that the first step in our presented analysis should have been to show the OLR-KWMF connection. We have determined a useful diagnostic of convective coverage upstream of the sounding site(s) that, once above a given threshold, allows us to find events for which the KWMF significantly rises above the background values. We then take these events and compare those with the strongest and those with the weakest MJO signal. From this, we believe we have found a meaningful signal in KWMF from the MJO. Perhaps key to this was ensuring that convection is active during the MJO events we analyze.

We believe that this is a sensible step in the knowledge of the field. If convection leads to stratospheric KWMF, then the MJO – the dominant pattern of intraseasonal variance in tropical convection – should have a discernable signal in stratospheric KWMF. While we strongly doubt that this idea is new to the field, there are not published studies about this to our knowledge. The work of Kiladis et al. (2005) does show penetration into the stratosphere of Kelvin wave-like perturbations during the MJO. This final bit of work was an attempt to begin connecting those dots. With the new means of analysis contained in the Discussion section, we hope that our attempts to make this tie are less abruptly introduced to the reader.

7) Like I said in the beginning, add some sentences or paragraphs highlight the uniqueness and novelty of your work.

R: we have added extra text in the introduction and summary noting that our long-term record of data and our application of short-time Fourier transforms are both novel. Our resolution experiments are novel as well, but drawing explicit attention to this seems inelegant where it would be in context.

---

## Author Response (AR2)

"Intraseasonal to interannual variability of Kelvin wave momentum fluxes as derived from high-resolution radiosonde data"
ACP-2016-1088

P6 L14: You may not require here the zonal phase speed to be positive. It has been already guaranteed in Eq. (3) with the positive and negative signs of k and m, respectively, resulting in the positive intrinsic frequency and thus positive intrinsic phase speed. Please remove this sentence or place it around P5 L5, providing why the m is defined as negative.

We move this sentence up in this paragraph, but change it to note that the requirement is satisfied by definition.

(If the authors follow the above comment) P6 L16–18: "Anywhere the above constraints are ..." : If the constraint for the positive intrinsic phase speed would not actually constrain the estimation (see the above comment), now it seems that you have only one constraint: Lx > 500 km. If so, the sentences in this paragraph could follow the last sentences in the previous paragraph (" ... 500 km."), after slight modification of the sentence in L16.

This is the case, so we make these changes and move a few sentences around to make these statements properly grouped.

P10 L7: I think that "is westerly. In easterlies" should be changed to "is westerly or strong easterly (|u| > N/|m|). In weak easterlies (|u| < N/|m|)", given that M ~ k/|m| ~ omega / (U|m| + N). Also, I do not understand the last two sentences in this paragraph ("i.e., a given ..."). Regarding the former sentence of the two, do you mean that "The increase by the overestimation of vertical wavelength in westerly and strong easterly dominates over the decrease by that in weak easterly"?

From this comment, we re-checked our analytic calculations and found that there was an error in the method. The effect is not reversed for easterlies, but is small relative to the effect for westerlies. Your comment is correct, except we find that even in weak easterlies, there is overestimation. This correction allows us to simplify the sentences in a way that is clearer. We also needed to rewrite a few sentences dealing with this subject in the Summary. These changes are not key to our findings.

P14 L20: "Our results may then give an upper estimate of the flux amplitude.": maybe true at the

altitudes relatively close to the tropospheric source. Around z = 23 ~ 25 km, the amplitude maxima could possibly propagate away to the east of these sites (i.e., on the Pacific ocean).

We adjusted the text to indicate that this upper estimate is valid perhaps for ~18-21 km.

P21 L10: "even if the whole of the Tropics is considered for determination of events" : Did the "all points between 10S-10N" mentioned in P18 L22 mean the whole longitudes ? Not the 30 degree longitude width in 10S-10N ? Anyway, I think the composite analysis with the cases of convections west of the observation site and that with the convection cases in whole tropics are totally different approaches, regardless of the qualitatively similar results from them.

In the latter analysis, the captured events may represent strong convection cases not only at the upstream but also downstream of the observation site. It may be not an expected result that the feature in Fig. 10 -- re. strong/weak MJO -- qualitatively holds also for the latter analysis, especially at the lowermost altitudes (z ~ 18 km) close to the convections.

We agree that this is a very different analysis. As such, we have removed mention of analyses over the whole Tropics. Sentences had be slightly adjusted to flow properly.

P3 L12: "positive contribution": Please rephrase this.

We have rewritten this sentence to be more direct.

P4 L17: Please change "for changes" to "for moderate changes". Otherwise, this sentence conflicts with the next sentence.

Done.

P5 L13: Please delete either of "reach or" or "or extend beyond", as the two have the same meaning here.

Done.

P6 L7: I would suggest changing "For positive omega and sufficiently strong easterlies," to "For sufficiently strong easterlies, if omega is set to positive,"

Done.

Figure 1b: The sign of zonal wind perturbations (contour) is not distinguishable, although one can find it from Fig. 1a. Please add at least one contour for each sign in Fig. 1b.

We adjusted the contour spacing to be 2 m s-1, which allows for distinction.

Figure 2a: The standard error should be defined in the figure caption.

It is now defined and we adjusted some text regarding this.

P10 L6: "verifies"

Done.

P16 L10: "the annual mean climatology displays descending signals of the flux and of easterlies during this span": Is this result previously mentioned with Fig. 6 ? If not, please include it there.

Mentioned now in the discussion of Fig. 6.

P18 L5: include a comma before "making".

Done.

P18 L12: Please delete "simply".

Done.

P21 L4–5: Please rewrite this sentence.

Rewritten.

P21 L9: "incipient"

Corrected.

P21 L16: "showed" ; "discernible"

Corrected.

P22 L11: Now Fig. 6 has been changed from that in the original manuscript. Please change "westerly" to "mainly westerly below z ~ 21 km".

Done.

P22 L26: Please delete "an annual periodicity with".

Done.

[revised manuscript text omitted]